

# Evaluation of the atmosphere-land-ocean-sea ice interface processes in the Regional Arctic System Model Version 1 (RASM1) using local and globally gridded observations

Michael A. Brunke[1], John J. Cassano[2], Nicholas Dawson[3], Alice K. DuVivier[4], William J. Gutowski,
Jr.[5], Joseph Hamman[4], Wieslaw Maslowski[7], Bart Nijssen[6], J. E. Jack Reeves Eyre[1], José C. Renteria[8],
Andrew Roberts[7], and Xubin Zeng[1]

[1]Department of Hydrology and Atmospheric Sciences, The University of Arizona, Tucson, AZ 85719, USA
[2]Cooperative Institute for Research in Environmental Sciences and Department of Atmospheric and Oceanic Sciences,
University of Colorado, Boulder, CO 80309, USA
[3]Idaho Power, Boise, ID 83702, USA
[4]National Center for Atmospheric Research, Boulder, CO 80305, USA
[5]Department of Geological and Atmospheric Sciences, Iowa State University, Ames, IA 50011, USA
[6]Department of Civil and Environmental Engineering, University of Washington, Seattle, WA 98195, USA
[7]Department of Oceanography, Naval Postgraduate School, Monterey, CA 93943, USA
[8]U.S. Department of Defense, High Performance Computing Modernization Program, Lorton, VA 22079, USA

*Correspondence to*: Michael A. Brunke (brunke@email.arizona.edu)

**Abstract.** The Regional Arctic System Model version 1 (RASM1) has been developed to provide high-resolution simulations of the Arctic atmosphere-ocean-sea ice-land system. Here, we provide a baseline for the capability of RASM to simulate interface processes by comparing retrospective simulations from RASM1 for 1990-2014 with the Community Earth System Model version 1 (CESM1) and the spread across three recent reanalyses. Evaluations of surface and 2-m air temperature, surface radiative and turbulent fluxes, precipitation, and snow depth in the various models and reanalyses are performed using global and regional datasets and a variety of in situ datasets, including flux towers over land, ship cruises over oceans, and a field experiment over sea ice. These evaluations reveal that RASM1 simulates precipitation that is similar to CESM1, reanalyses, and satellite-gauge combined precipitation datasets over all river basins within the RASM domain. The possible reasons for this result are discussed. Snow depth in RASM is closer to upscaled surface observations over a flatter region than in more mountainous terrain in Alaska. The sea ice interface is well simulated in regards to radiation fluxes which generally fall within observational uncertainty. RASM1 surface temperature and radiation biases are shown to be due to biases in the simulated mean diurnal cycle. Development of RASM2 aims to address these biases.

## 1 Introduction

The late 20[th] and early 21[st] centuries have been marked by dramatic changes in the northern high latitudes. Most notably was the rapid decline in sea ice cover (e.g., Serreze et al., 2007; Comiso and Hall, 2014), that accelerated during the first



decade of the 21$^{st}$ century (e.g., Comiso et al., 2008; Stroeve et al., 2012; Swart et al., 2015). Since then, sea ice extent partially recovered in 2013-2015 (Swart et al., 2015), followed by further declines in 2016-2017 (http://nsidc.org/arcticseaicenews). Sea ice thickness also decreased along with the sea ice extent decline (Johanssen et al., 2004; Serreze et al., 2007). This reduced sea ice extent decreaces the surface albedo, initiating a positive feedback in which

the surface is warmed by an increase in absorbed solar radiation. This further enhances sea ice melt (Hartmann, 1994) by producing more first-year sea ice, which is thinner and easier to melt in spring (Stroeve et al., 2012). This positive feedback causes warming to be highest in the Arctic, a process which has been termed Arctic amplification (Holland and Bitz, 2003; Johanssen et al., 2004; Serreze and Francis, 2006; Serreze et al., 2009). Further enhancement of Arctic warming is realized with increased water vapor, a greenhouse gas, from more evaporation over the additional open water (Screen and Simmonds,

2010). Even more warming occurs from the large reductions in snow cover over land (Estilow et al., 2015) which reduces the surface albedo during winter (Serreze et al., 2009; Comiso and Hall, 2014). Also, permafrost is thawing, which may release substantial portions of the large amount of carbon stored underground to the atmosphere (Schuur et al., 2015; Lawrence et al., 2015). This may further enhance warming in the Arctic.

Because of the region's increased sensitivity to global warming, the Arctic is an important region for GCMs and ESMs to
model correctly. Yet even though GCMs and ESMs capture the general large-scale and long-term temperature trends in the Arctic, they have difficulty capturing other climatic trends in the region (Serreze and Francis, 2006). For instance, while these models generally simulate the overall decline in sea ice extent and area, there is a large spread in the simulated sea ice decline among the various models (Stroeve et al., 2007; Zhang and Walsh, 2006) and many fail to capture the recent acceleration in that decline (Stroeve et al., 2007; Zhang, 2010). Such biases lead to a large range in the simulated polar
amplification from these models due to variations in the sea ice state caused by differences in the representation of physical processes (Holland and Bitz, 2003) and due to errors in simulated atmospheric circulation (Maslowski et al., 2012; DeRepentigny et al. 2016). The latter is partly due to errors in the phase of the Arctic Oscillation and North Atlantic Oscillation (Moritz et al., 2002; Stroeve et al., 2007) of which they are not expected to portray accurately.

The improvement of GCMs and ESMs in the Arctic may be facilitated by an Arctic regional system model as was proposed
by Roberts et al. (2010). Such a regional model would provide a stepping stone toward the development of high resolution fully coupled global models with sophisticated polar representations. Many physical and biogeochemical processes in the Arctic are contingent upon interfacial exchanges at fine spatial scales and short time scales and cannot be represented within the computational constraints of the current generation of ESMs (Roberts et al., 2011). The development of such a new regional coupled model, the Regional Arctic System Model (RASM) presented here, incorporates high-resolution
atmosphere, ocean, sea ice, and land surface components, and accommodates expansion to mountain glaciers, ice sheets, dynamic vegetation, and biogeochemistry modules (Maslowski et al., 2012). The first version of RASM (RASM1) incorporates WRF as the atmospheric model, the Variable Infiltration Capacity (VIC) land surface model, a streamflow routing model (RVIC), the Parallel Ocean Program (POP) ocean model, and the Los Alamos Community Sea Ice Model (CICE). The latter two are also used in the global Community Earth System Model (CESM), and the development of RASM



has contributed to refinements in the CICE version 5 (Hunke et al., 2015). Along with the use of CESM's ocean and sea ice models, coupling between the various components is performed by the CESM coupler, CPL7 (Craig et al., 2012, http://www.cesm.ucar.edu/models/ccsm4.0/cpl7/), modified for regional modeling (Roberts et al., 2015).

The development of the version of WRF used in RASM for long-term climate simulations for a pan-Arctic domain (Cassano

et al., 2011) was motivated by the adaptation of WRF for polar applications (Polar WRF, Hines and Bromwich, 2008; Bromwich et al., 2009), which is being used to produce the Arctic System Reanalysis (ASR, Bromwich et al., 2016). This grew out of the previous development of Polar MM5 (Bromwich et al., 2001; Cassano et al., 2001). In developing RASM1, lessons were also heeded from the existing lineage of Arctic-centric models like the Arctic Climate System Model (ARCSyM, Lynch et al., 1995; Lynch et al., 1998; Lynch and Cullather, 2000; Lynch et al., 2001) and the coupled ocean-

atmosphere models of the Rossby Centre Atmosphere-Ocean RCM (RCAO, Döscher et al., 2002, 2010) and HIRHAM coupled to the North Atlantic-Arctic ocean-sea ice model (NAOSIM) or the Modular Ocean Model (MOM) (Dorn et al., 2007; Rinke et al., 2003).

RASM1 and its simulations evaluated here are described in more detail in Section 2a. These or similar simulations have also been evaluated in Hamman et al. (2016, 2017) and Cassano et al. (2017). The former focused exclusively on the land

surface climatology and hydrology, and the latter compared the near-surface atmospheric climate in RASM to a single reanalysis. What is presented here is an evaluation of the capability of these simulations in regards to atmosphere-land-ocean-sea ice interface processes by comparing with observational data and using three reanalyses and an ESM as baselines for the performance of RASM1. It should be noted that it is not the goal of RASM1 to be always comparable to the ESM and reanalyses, as these may not always compare well with the observational data. Instead, RASM1 should be better than

the ESM for quantities that the ESM does not simulate well and should be comparable for quantities that the ESM simulates well. The focus here is on evaluating RASM and providing pathways for improving this particular model which will be a useful tool for gaining improved understanding of the Arctic climate system. The ESM used here, CESM1, is also described in Section 2a, and the reanalyses used are described in Section 2b. The observational data, both globally gridded data and surface observations are described in Sections 2c and d, respectively. The evaluation is given in Section 3. Finally,

conclusions are given in Section 4.

**2 Model simulations and evaluations**



### 2.1 Model simulations and evaluation datasets

#### 2.1.1 The Regional Arctic System Model (RASM)

RASM is run over a pan-Arctic domain that encompasses the entire Arctic Ocean and the surrounding river basins (Fig. 1). The atmosphere and land models are run on the same ~50 km polar stereographic grid, while the ocean and sea ice models

are on the same 1/12° (~9 km) rotated sphere grid.

RASM includes version 3.2 of the Advanced Research WRF (Skamarock et al., 2008) modified for use in the Arctic (Cassano et al., 2011; Cassano et al., 2017). In order to successfully couple within RASM, WRF's boundary layer, surface layer, and radiation parameterizations have been adapted. Details of these changes and other information on the WRF configuration used in RASM can be found in DuVivier et al. (2015) and Cassano et al. (2017). Of particular relevance to

this paper is the use of spectral nudging to reduce biases in the regional simulation (Glisan et al., 2012; Cassano et al., 2011). The nudging of temperature and winds are ramped upwards from zero starting at ~540 hPa and a horizontal scale of ~3400 km with reanalysis fields. This nudging constrains only the large-scale circulation above the boundary layer (Cassano et al., 2017). Nudging has been found to mostly affect sea level pressure biases (Berg et al., 2013) but has been found not to impact the climatology of surface quantities and interactions between model components that are of particular interest to this

study (Berg et al., 2013, 2016; Cassano et al., 2011; Glisan et al., 2012). Instead, what is important to model biases are errors in the model physics. Nudging would also have a minimal impact at the highest parts of the Greenland ice sheet (at ~650 hPa), since it starts from zero at a higher altitude at ~540 hPa.

Version 4.04 of the land model VIC (Liang et al., 1994, 1996) used in RASM is modified for coupling to the other components and to include a broadband snow albedo that depends on vegetation cover (Barlage et al., 2005). Other

modifications include an increase in the bare surface albedo to simulate bare land ice at very high latitudes, and a decrease in land surface emissivity throughout the region to 0.97 to be consistent with the other components. Hamman et al. (2016) describe this version of VIC in more detail. RASM also includes a model to route streamflow from the land to the river outlets into the ocean. This river routing model, RVIC, is described in more detail in Hamman et al. (2017).

RASM uses version 2 of the ocean model POP (Smith et al., 1992; Dukowicz, 1994; Smith et al., 2010) modified for a

regional closed boundary domain on a 1/12° (~9 km) rotated sphere grid demonstrated in Fig. 1. Climatological sea surface temperatures (SSTs) provide lower boundary conditions to the part of the WRF domain beyond this regional ocean model domain. The boundary conditions for this regional version of POP are provided by the monthly Polar Science Center Hydrographic Climatology (PHC) temperature and salinity climatology interpolated to model time steps. The oceanic state within the first 71 grid cells from the ocean model boundary undergoes Newtonian relaxation for all model layers such that

the relaxation strength is 30 days for the first 48 grid cells and linearly decreases to 0 at 71 grid cells (Roberts et al., 2015). POP is coupled to the sea ice model, CICE, using methods described in Roberts et al. (2015). Since then, RASM has incorporated a newer version (version 5; Hunke et al., 2015) of CICE. (The CESM simulations described later use version 4 of this sea ice model.) This later version of CICE has been configured in RASM with anisotropic sea ice mechanics



(Tsamados et al., 2013) and explicit level-ice melt ponds (Hunke et al., 2013). The latest baseline RASM simulation presented in this paper uses mushy-layer sea ice thermodynamics of Turner and Hunke (2015), which incorporates a prognostic salinity profile and uses the associated liquidus relation to calculate a salinity-dependent freezing temperature at the ice-water interface.

The current baseline simulation, RASM1, is as described above using the MYNN (Nakanishi and Niino, 2006) boundary layer and Kain-Fritsch (Kain, 2004) convection schemes in WRF. The MYNN and Kain-Fritsch schemes were found to produce a more realistic boundary layer height, liquid water path, and downward shortwave radiation in stratocumulus (Jousse et al., 2016) such as those prevalent over the sub-polar oceans. This is very similar to the "RASM_atm_ice" simulation assessed in Cassano et al. (2017) with improvements to the ice-ocean coupling. The ice-ocean coupling

improvements mostly affected sea surface temperature and salinity but had a minimal impact on sea ice concentration or thickness (not shown).

The initial conditions for POP and CICE were provided from a spin-up using CORE-2 forcing and runoff (Large and Yeager, 2009) from 1948, and those for VIC from a spin-up from January 1948 to August 1979 using the forcing dataset of Sheffield et al. (2006). The European Centre for Medium-Range Weather Forecasts (ECMWF) interim reanalysis (ERA-

Interim; Dee et al., 2011) was used as lateral boundary conditions for the atmospheric model and to nudge the upper atmosphere of the model. The Climate Forecast System Reanalysis (CFSR, Saha et al., 2010) was also used for lateral atmosphere model boundary conditions and to nudge the model upper atmosphere (while continuing to use the PHC climatology for the ocean model boundary conditions) producing results that are generally not significantly different from those using ERA-Interim over most of the domain (supporting information Fig. S1). The differences along the edge of the

domain are produced by differences in the boundary conditions.

The RASM1 simulation was run fully coupled for 1979-2014. The period 1979-1989 is not analyzed here, as the ocean and sea ice needed to relax into its climatological state. For instance, while domain average sea surface temperature (SST) is stable throughout the simulation, sea surface salinity slightly decreased from 1979 into the 1980s in RASM1 (supporting information Fig. S2). Thus, analysis is made to results from 1990 onwards, generally focusing on the period up to 2009 to

have a consistent comparison with the period available that all of the reanalyses used here.

### 2.1.2 The Community Earth System Model (CESM)

To provide a baseline for the capability of RASM1 in simulating interface processes, we compare the climate from RASM1 to that of CESM, the modeling system from which portions of RASM were branched. Output from the 30-member CESM large ensemble (LE) (Kay et al., 2015) is used here, since the CESM-LE output in the NCAR database includes 6-hourly,

daily and monthly means of many of the quantities investigated here. We refer to CESM-LE as CESM1 henceforth and use output from 1990 to the end of the simulations in 2005.





### 2.2 Reanalyses

To further evaluate RASM1 simulations, we compare them to the spread in the latest generation of reanalyses: the Modern Era Retrospective Analysis for Research and Applications, version 2 (MERRA-2, Gelaro et al., 2017), ERA-Interim (Dee et al., 2011), and National Centers for Environmental Prediction (NCEP) CFSR (Saha et al., 2010a). These reanalyses have
been shown to be the most consistent with independent observations in the Arctic (Lindsay et al. 2014). The last two have been used for lateral and internal boundary conditions for RASM with similar results (Fig. S2).

The temporal and horizontal resolutions of the reanalyses used in this study are summarized in Table S1 in the supporting information. MERRA-2 data used here include the surface turbulent flux, surface radiation, and single-level diagnostics data collections given at the model's native horizontal resolution of 0.5° latitude × 0.625° longitude. Hourly means, monthly
means, and monthly mean diurnal cycles are used here. Monthly mean ERA-Interim data and 3-hourly means derived from a combination of the surface analyses and forecasts are used here. These are at the model horizontal resolution of ~0.703° × 0.702°. The 3-hourly monthly mean diurnal cycles on a uniform horizontal grid of 0.75° × 0.75° are also used. CFSR's monthly mean (derived from the 0-5 h forecasts) and hourly time-series products are utilized here at the reanalysis model resolution of ~0.31° × 0.31°. For all reanalyses, we use data from 1990-2009 when data from all three reanalyses were
available.

### 2.3 Global evaluation datasets

The simulated monthly means are first evaluated using several global monthly mean gridded datasets. This is done by regridding the model and reanalysis data to the various product resolutions for comparison in Section 3a.

Monthly mean 2-m land surface air temperature (LSAT) is compared to the dataset generated by Wang and Zeng (2013,
hereafter WZ13). WZ13 includes adjusted hourly 2-m air temperature on a 0.5° × 0.5° horizontal grid individually for four reanalyses: MERRA, ERA-Interim, the ECMWF 40-year reanalysis, and the NCEP-NCAR reanalysis. The adjustments include the downscaling to the 0.5° × 0.5° grid, temporal interpolation to hourly time resolution, and correction of the reanalysis monthly mean maximum and minimum air temperature biases according to the University of East Anglia's Climate Research Unit (CRU) surface temperature data (New et al., 2002; Osborn and Jones, 2014). WZ13 show that the
reanalyses are much more consistent with each other after the adjustments, eliminating large spurious jumps seen in the individual reanalysis regional means. In addition, the monthly mean land SAT diurnal range derived from averaging the hourly values is more reflective of the diurnal effects than the monthly mean diurnal range in Arctic winter (Wang and Zeng, 2014). However, WZ13 acknowledged that their adjustment was possibly problematic over Greenland due to the use of biased CRU data which was confirmed by Reeves Eyre and Zeng (2017). In this study, we utilize only the adjusted air
temperatures from the two newer reanalyses (MERRA and ERA-Interim), taking the average of the two for 1990-2009.

Sea surface temperature (SST) is evaluated using version 3 of the Hadley Centre SST (HadSST3.1.1.0; Kennedy et al., 2011a,b) dataset on a 5° × 5° horizontal grid. This dataset is not globally complete but most ocean grid cells within the





RASM domain contain data. The actual monthly mean SSTs for 1990-2009 are derived from the anomalies by adding the climatological mean SSTs. The range of uncertainties due to various biases is considered in the development of this dataset (Kennedy et al. 2011b). The standard deviation of these uncertainties is no more than 0.43°C within the RASM domain.

Sea ice concentration and extent are important quantities to be assessed in such a regional climate model for the Arctic.
These were preliminarily evaluated in Cassano et al. (2017) and will be more thoroughly evaluated in a subsequent paper about CICE as used in RASM. Still, we will briefly assess this to understand some of the model biases over and around the margins of the sea ice through use of the National Oceanic and Atmospheric Administration (NOAA) climatic data record (CDR) sea ice concentration product (Peng et al., 2013; Meier et al., 2013, 2014).

To understand the biases in 2-m air temperature or surface temperature, we evaluate the surface energy balance in the
models. Surface radiation is evaluated using the measurements from the Clouds and the Earth's Radiant Energy System (CERES) satellite for 2001-2009. CERES's level 3B Energy Balanced and Filled (EBAF)-Surface (Li et al., 1993; Li and Kratz, 1997; Gupta et al., 1997) provides surface radiative fluxes on a $1° \times 1°$ global horizontal grid. The downward incident shortwave and longwave radiative fluxes were shown to have root-mean-square differences of 13.3 and 7.1 W m$^{-2}$ over land and 7.8 and 7.6 W m$^{-2}$ over ocean when compared to 10 years of in situ observations (Kato et al., 2013).

Finally, we use NCEP's Climate Prediction Center Merged Analysis of Precipitation (CMAP, Xie and Arkin, 1997) to evaluate precipitation over the period 1990-2009. Monthly mean values on a $2.5° \times 2.5°$ grid are derived from merging gauge observations, estimates from several satellites, and data from the NCEP-NCAR reanalysis (Xie and Arkin, 1997). This was preferred over the similar Global Precipitation Climatology Project (GPCP) dataset (Adler et al., 2003), which was found to have a worse depiction of monthly precipitation than reanalyses in the Arctic (Serreze et al., 2005). Furthermore,
Adler et al. (2012) pointed out that GPCP was most biased at high latitudes in winter. Over the northern high latitudes, this was found to be due to an overestimate in snowfall over northwestern Eurasia (Behrangi et al., 2016).

### 2.4 Local surface observations

We use point observations to further evaluate RASM. First, we use observations of 2-m surface air temperature (SAT) from five automated weather stations (supporting information Table S2 and the map in the lower right of Fig. 11) from the
Greenland Climate Network (GC-Net) on the Greenland Ice Sheet. These stations have been operational since the 1990s (Steffen and Box, 2001), and the five chosen for this study have some of the longest records in the accumulation zone of the ice sheet above 2300 m. We compare the SAT observations from these stations distinctly with the individual model or reanalysis grid cell values containing these stations.

Over land elsewhere, we use tower observations from FLUXNET (Baldocchi et al., 2001), a global network of more than
100 locations where fluxes of $CO_2$, water, and energy are measured at various heights above the surface. In this study, we use observations of 2-m air temperature, sensible heat flux, latent heat flux, downward shortwave radiation, net total radiation, 10-m wind speed, and precipitation rate from 26 high-latitude sites across North America and Eurasia listed in





supporting information Table S3. These locations were chosen, because they have at least three years of data during the evaluation period of 1990-2009 with the exception of US-HVa which only has five months of data during the summer of 1994. Despite the very short observation record of US-HVa, we use it here, as it with CA-Man was used to evaluate RASM1a in Hamman et al. (2016). Additionally, CA-Man and seven other flux towers (NS-1 through 7) happen to be

clustered within one RASM grid cell. We compare the mean observations from these eight towers to that of the model or reanalysis grid cell containing these towers. Similarly, the observations from CA-Man and the other 18 towers (i.e., all except NS-1 through 7) are compared to the individual model or reanalysis grid cell values containing these towers.

RASM snow depth over land is evaluated with upscaled in situ observations using the methodology of Dawson et al. (2016). The upscaling within $2^{o} \times 2^{o}$ boxes is performed by a piecewise linear regression of 100-m elevation bands. This method

was found to compare better to observations than other upscaling methods, including inverse distance squared weighting, optimal interpolation, and kriging. Also, the area averages of these upscaled observations compare well to National Weather Service Snow Data Assimilation System (SNODAS) over both mountainous and flat boxes (Dawson et al., 2016). With our focus on the Central Arctic, two $2^{o} \times 2^{o}$ boxes (Fig. 1) were selected for representing relatively flat land (ALASKA MID, with a mean elevation of 525 m, range of 1389 m, and standard deviation of 266 m) and relatively mountainous land

(ALASKA SOUTH, mean elevation of 513 m, range of 2376 m, and standard deviation of 509 m). Each of these boxes includes observations from at least four locations per day over the periods 2010-2014 and 2008-2014, respectively. The daily averages of all RASM grid cells within each box are compared to the daily area averages of the upscaled data.

Over sea ice, we use meteorological and flux observations from the Surface Heat Budget of the Arctic (SHEBA, Uttal et al., 2002; Persson et al., 2002) between October 1997 and October 1998. These include measurements made at the 20-m tower

at the main camp and from four Portable Automated Mesonet (PAM, Militzer et al., 1995) stations surrounding the main camp. On the tower, measurements were made at several levels. Here, we use the sensible heat fluxes derived from fast measurements of temperature and wind made by sonic thermometers and anemometers and latent heat fluxes derived from measurements from a fast hygrometer at 8.1 m. Upward and downward shortwave and longwave radiation were measured by pyranometers and pyrgeometers on nearby masts at 1.5-2 m height. Surface temperature was measured nearby by a

downward pointing radiation thermometer. At the PAM stations, we use sensible heat fluxes, surface radiation, surface temperature, and near-surface air temperature made using similar measurements. Further discussion of these instruments and their uncertainties is provided by Brunke et al. (2006) and Persson et al. (2002). We compare the average of the tower and PAM stations with the values from the model or reanalysis grid cell containing the combined observations at the corresponding day.

Over ocean, we use flux and meteorological observations made aboard ships in three field campaigns that fall within the RASM domain: the Fronts and Atlantic Strom Track Experiment (FASTEX) from December 1996-January 1997 followed by Couplage avec l'Atmosphère en Conditions Hivernales (CATCH) from January-February 1997 in the North Atlantic and the National Oceanic and Atmospheric Administration's cruise to service its moorings in the North Pacific (Moorings '99) in September and October 1999 (Fig. 1). We use the eddy covariance latent and sensible heat fluxes from the U.S. cruises



(FASTEX and Moorings '99), while only inertial-dissipation fluxes were available for CATCH. Flow distortion, ship motions, and environmental conditions were accounted for as in Brunke et al. (2003). We only use observations deemed far enough within the active ocean domain (Fig. 1). Still, the location of the CATCH and FASTEX observations used are close enough to the edge of the active domain for the model state and fluxes to be influenced by the boundary conditions. We still use them because of the lack of high-latitude ocean observations. We compare the daily averages of the cruise data to the daily mean model or reanalysis grid cell value containing the daily average observations.

In this study, latent and sensible heat fluxes are considered positive in the upwards direction. The magnitude of the radiation components are considered (i.e., always positive) such that a net radiative flux $R_{net} = R_{down} - R_{up}$ where $R_{up}$ is the upward flux and $R_{down}$ is the downward flux. Thus, the net radiative fluxes are considered positive downward into the surface.

## 3 Results

### 3.1 Domain-wide and regional comparisons

We first evaluate RASM1 across the pan-Arctic domain for the period 1990-2009 (2001-2009 for CERES). In Fig. 3, the biases in RASM1's simulated precipitation relative to CMAP (the mean values of which are presented in supporting information Fig. S3 for reference) are compared to those of ERA-Interim and CESM1 in January and July. We pick these months to represent snow-covered and relatively snow-free periods, respectively, over most of the domain. We focus on biases poleward of 50°N, because these influence the simulation of the Arctic Ocean. RASM1 precipitation biases are very similar to those in ERA-Interim, which is representative of the biases from the other two reanalyses, in both January and July (Fig. 2a-d). This includes the high overestimates of precipitation (as much as > 4 mm day$^{-1}$) over both subpolar (North Pacific and Atlantic) ocean basins. CESM1 also overestimates precipitation over the subpolar oceans in January, whereas it is only slightly overestimated over land (Fig. 2e). The mean bias across the regional river basins in January is 0.01, 0.10, and 0.12 mm day$^{-1}$ in RASM1, ERA-Interim, and CESM1, respectively. In July, CESM1 is similarly biased to RASM1 and the reanalyses across these basins in July (Fig. 2c,d,f) with mean biases of 0.30, 0.83, and 0.30 mm day$^{-1}$ in CESM1, RASM1, and ERA-Interim, respectively. The biases relative to GPCP are generally of the opposite sign from CMAP.

This is further illustrated by the mean annual cycle averaged over the Ob River basin indicated by the brown region in Fig. 3a. This basin is representative of all of the river basins within the domain except the Amur (the red region in Fig. 3a). RASM1's precipitation rate lies between that of CMAP and GPCP before May and after July and is within the spread in the reanalyses from January to May (with the exception of March). Additionally, RASM1 precipitation is within CESM1's ensemble spread for every month except in June. This suggests that RASM1 simulates precipitation fairly well in the Ob River and other similar basins. In the Amur basin, GPCP and CMAP are more consistent with each other. GPCP in this basin is at or near the bottom of the reanalysis spread up until August. RASM1 precipitation here is lower than GPCP and CMAP throughout the year and only barely within the CESM1 ensemble spread in March and April.



The similarity of precipitation in RASM1 to that of ERA-Interim could be due to the lateral boundary conditions (BCs) or to the spectral nudging imposed from the reanalysis. Using a different reanalysis for BC and nudging could produce different simulated precipitation. To test the impact of the choice of reanalysis used for the BCs and for spectral nudging, we compare the RASM1 run using ERA-Interim BCs with that using CFSR BCs. Fig. S2 in the supporting information shows

5 that the precipitation differences between RASM1 using ERA-Interim BCs and CFSR BCs is minimal (differences of < 1 mm day$^{-1}$ in January and July with differences of ≥ 1 mm day$^{-1}$ in only a few isolated regions in July) and statistically significant essentially only near the domain boundaries. Admittedly, the CFSR precipitation biases are similar to ERA-Interim's (not shown); this may explain why precipitation in both simulations is so similar. RASM1 with ERA-Interim BCs is more strongly correlated to ERA-Interim precipitation than to that of CMAP, but correlations of 0.9 or more are found

10 only in isolated regions off of the west coasts of North America and Europe (not shown).

In contrast, surface temperatures were shown to have large biases in a very similar version of RASM that preceded RASM1 (referred to as RASM_atm_ice) when compared to ERA-Interim in Cassano et al. (2017). The land SAT biases are further substantiated here by comparing to WZ13 in January and July in Fig. 4. For reference, the mean values in WZ13 for these two months are shown in supporting information Fig. S4. Besides over Greenland, RASM1 land SAT in January is also

15 much colder over the low-lying land areas with the coldest biases in northern European Russia (NRU, the dark blue box bordered by 60°-75°N, 30°-90°E) just south of the Kara Sea (Fig. 4a). The mean bias in this region in January is -8.24°C. Similar wintertime cold biases are simulated in RASM1 compared to reanalyses over the nearby central Arctic (defined as everywhere 70°N and poleward) as well (supporting information Fig. S5). July SAT, on the other hand, is biased high over much of the land within the domain. These model biases can be placed into context by comparing to the reanalyses. The

20 magnitude of land SAT biases in both January and July are generally much lower in ERA-Interim (Fig. 4b,c) and MERRA (not shown). CFSR's January land SAT biases are much higher than the other reanalyses or RASM1, while July biases are similar to ERA-Interim's (Fig. 4e,f).

Reeves Eyre and Zeng (2017) have shown that CRU (and hence WZ13) is biased high in winter compared to Greenland automatic weather stations. Here, interior SAT relies on interpolating data from the few coastal stations vertically to the top

25 of the ice sheet. Therefore, RASM1 and the reanalyses appear to be colder than WZ13 in January over most of Greenland. A warm bias for related reasons over areas of higher terrain elsewhere in the domain results from WZ13 being too cold. Thus, RASM1 and the reanalyses appear to be too warm when compared to WZ13. These biases are further discussed below.

We return to the cold biases over the flatter terrain of the domain. The mean annual cycle in land SAT for all land in NRU

30 where the coldest biases are in January (Fig. 4a) is given in Fig. 5a-c. The cold bias in RASM1 from WZ13 is clearly evident in this region in winter and fall, while this model's SAT is too warm in June and July. The reanalysis spread indicated by the gray shading tightly surrounds the WZ13 mean throughout the year; RASM1 is outside of this for most of the year. The CESM1 ensemble mean is also too cold in winter. We can further evaluate RASM1's biases by comparing



with the spread (minimum to maximum) of the CESM1 ensemble member means. RASM1 is within the CESM1 ensemble spread only in March, April, August, and December.

Cassano et al. (2017) suggested that the SAT biases in RASM1 are the result of cloud errors as evidenced by surface incident downward radiation biases. In winter, downward incident shortwave (SW) radiation is near-zero in NRU (Fig. 5b).

Downward incident longwave (LW) radiation is much more substantial (~207 W m$^{-2}$ in January in CERES) and, therefore, more important to the surface energy balance at this time of year (Fig. 5c). RASM1 downward incident LW radiation is ~30 W m$^{-2}$ lower than CERES (which is lower than the reanalyses) in January and February. Also, RASM1's biases are lower than the CESM1 ensemble spread at this time. For instance, RASM1 is ~17 W m$^{-2}$ lower than the CESM1 ensemble minimum in February. RASM1 downward incident LW radiation is within the reanalysis spread from May to September

and within the CESM1 ensemble spread from March until the end of the year despite being slightly lower than CERES throughout the year (Fig. 5c).

RASM1 downward incident SW radiation in NRU is much higher in summer with a maximum of 298 W m$^{-2}$ versus 224 W m$^{-2}$ in CERES. CESM1 is also biased high in summer with the annual maximum of the ensemble mean being 285 W m$^{-2}$, allowing the ensemble spread to be above CERES from May to the end of the year. Again, the reanalysis spread surrounds

CERES throughout most of the year. As suggested by Cassano et al. (2017), too little downward incident LW radiation in winter would be the result of too little or too optically thin cloud (possibly from there being not enough supercooled water in mixed-phase clouds) being simulated. More cloud or optically thicker cloud would direct more LW radiation to the surface. Too much downward incident SW radiation in summer would also be resultant of too little or too optically thin cloud, as more SW radiation would be reflected before reaching the surface if more or thicker clouds were simulated. Upward

incident SW radiation over this region is also overestimated (not shown), consistent with the overestimated downward SW radiation produced by these cloud errors. This is further substantiated by the downward incident LW radiation that continues to be too low compared to CERES in summer as well. As with earlier RASM simulations, cloud variables were unable to be included in the model output, so we cannot further substantiate this by evaluating the simulated clouds. This will be a focus of RASM2 which will include cloud variables.

We further illustrate WZ13's wintertime cold bias over higher terrain outside of Greenland by looking at the mean annual cycle shown in Fig. 5d-f for the northeastern Siberia (NSIB) region demarcated by the red box (60°-75°N, 30°-90°E) in Fig. 4a. This region encompasses the apparent January warm biases in the mountainous terrain of NSIB, and thus the regional mean RASM1 SATs are slightly higher than WZ13 (Fig. 5d) as would be expected from Fig. 4. However, RASM1 downward incident radiation biases are similar here to those in NRU (Fig. 5e,f) as are the latent and sensible heat fluxes (not

shown). In these mountainous regions, CRU (and thus WZ13) is probably biased because of the limited number of observational sites mostly located in the valleys, so that temperatures have to be interpolated vertically. This interpolation is further complicated by the prevalence of surface temperature inversions in winter.

RASM1 SSTs are compared to HadSST in Fig. 6a,b. There are large differences in SST in the marginal ice zones, including the largest biases (in excess of -14°C) in Fram Strait, due to differences in sea ice extent (Cassano et al., 2017). In particular,



there is much more sea ice in RASM1 in the Fram Strait and Greenland and Barents Seas than in the NOAA CDR product in both January and July (supporting information Fig. S6). RASM1 SST biases are mostly < -2°C over the rest of the open ocean in January (Fig. 6a) but can be > +2°C over parts of the sub-polar North Pacific and Atlantic in July (Fig. 6b).

**3.2 Comparison to land surface observations**

We now use in situ observations over land to further explore several of the biases discussed above. To substantiate the above comparisons of RASM1 with global reference data elsewhere over the land, we further compare the modeled surface meteorology, fluxes, and radiation to in situ observations made at the FLUXNET towers. There is some uncertainty in comparing point measurements to grid cell mean quantities from a model simulation or a reanalysis. We use a cluster of eight FLUXNET towers in northern Manitoba (CA-Man and CA-NS1 through 7) that happens to span a RASM1 grid cell

within the boreal forest as an example of the possible uncertainty of such a comparison. This is not the only cluster to do so, but most clusters cover only a small area with ~2 or 3 towers to sample vegetation diversity. The towers in this Manitoba cluster are more spread out throughout a RASM1 or typical reanalysis grid cell. Therefore, this would be a better sample to investigate uncertainty arising from evaluating grid cell means in the models to these point measurements, especially since the terrain here is relatively flat (tower elevation ranging from 245 to 291 m). CA-NS1 through CA-NS7 were only

operational from 2001 or 2002 to 2005, while CA-Man was a long-term site that was operational from 1994-2008 (supporting information Table S2). Fig. 7 shows that the CA-Man's SATs and net radiation annual cycles are very similar to the eight-tower mean throughout the year, while latent heat (LH) and sensible heat (SH) fluxes may be substantially different from the mean. This suggests that there is more uncertainty in using single-point measurements of turbulent fluxes than in SAT and net radiation over a region that is relatively flat.

We can use the range in tower observations to evaluate the RASM1 simulation. If the simulated value falls outside of this range, then the simulation might be problematic. Even with the large winter cold biases and warm biases in summer, SAT is generally within the observational spread in this region except in January, November, and December when it is below the observational minimum (Fig. 7a). Net radiation is also below the observational spread in these months, while it is above the observational maximum from June to August (Fig. 7b) with a maximum that is 33 W m$^{-2}$ higher than the cluster maximum.

RASM1 downward incident SW radiation is above the observational uncertainty from May to December (Fig. 7c). There is no downward incident LW radiation measured in this cluster, but we can infer from the downward SW radiation that the net radiation high bias in summer is due to the excessive downward incident SW radiation. However, the downward incident SW radiation biases are minimal in winter, so the negative net radiation biases in winter are likely due to downward incident LW radiation biases as seen in Fig. 5 and in Cassano et al. (2017). The downward incident SW radiation biases in summer

result in latent heat (LH) fluxes in RASM1 from March until September (Fig. 7d) that are much larger than observed with a maximum that is 68 W m$^{-2}$ higher than the cluster maximum. Sensible heat (SH) fluxes are in closer agreement with the observations, only being slightly below observational uncertainty in the first three months of the year and slightly above observational uncertainty from July to September (Fig. 7e). This analysis suggests that, for this region, wintertime SAT and




net radiation, summertime net and SW radiation, and spring and summertime LH flux biases is unlikely to be attributed to the uncertainty in comparing point measurements to grid scale mean simulated values.

Another measure of how well RASM1 simulates the mean annual cycle in these quantities is to compare it with the spread in the reanalyses. However, the reanalysis spread is not always within the observational spread here at this Manitoba cluster, and, in such cases when RASM1 is within the observation spread, it is better than the reanalyses. The reanalyses fall within the observational spread for SAT throughout the year, but not necessarily for radiation or SH and LH fluxes, quantities that are not assimilated. Thus, the RASM1 autumn and winter cold biases are also below the reanalysis spread, while simulated net radiation is within the reanalysis spread during this time. On the other hand, net and downward incident SW radiation is higher than the reanalysis spread in summer. Model LH and SH flux is above even the reanalysis spread during the summer maximum. However, the reanalysis spread is above the observational spread in autumn and winter, whereas RASM1 compares well with the near-zero observations at this time of year.

To evaluate how well RASM1 is performing across the domain, we look at the other single FLUXNET towers (Fig. 8). The model winter cold bias is evident at all locations, especially at the more northern sites. However, simulated SAT at the tundra sites are biased high from late winter into summer, while it is better simulated across boreal Canada and at the temperate stations (Fig. 8a).

The cold biases are generally associated with negative net radiation biases, and warm biases are generally associated with positive net radiation biases > 50 W m$^{-2}$ at a few locations (Fig. 8b). In winter, the negative winter net radiation biases are associated with downward LW biases, while downward incident SW biases dominate the positive net radiation biases in summer (Fig. 9a,b). These biases are not diminished by the upward radiation biases which are much smaller than the downward radiation biases (not shown). The LH and SH fluxes have large positive biases in spring to summer and minimal (magnitude < 5 W m$^{-2}$) or slightly negative biases in winter (Fig. 9c,d) that correspond to similar net radiation biases (Fig. 8b). These biases are consistent with what is seen at the full Manitoba cluster.

To further understand these monthly means, we analyze the monthly mean diurnal cycles at CA-Man and two other sites, one in the boreal forest of northern Europe (FI-Hyy) and another in the Alaskan tundra (US-Ivo) in Fig. 10. We focus on July when monthly mean SAT is biased quite highly positive in RASM1, but the mean diurnal cycle in SAT differs slightly between the three sites. For example at CA-Man, RASM1 SAT is biased low within the observational interannual variability (±1σ) at night and biased warmer than observed interannual variability and the spread in the reanalyses during the day (Fig. 10a). At FI-Hyy, the nighttime biases are more negative, and the daytime warm biases are not as high (Fig. 10b). Finally, at US-Ivo, the SAT is biased high throughout the day but are within the larger observational variability of ~3°C (Fig. 10c). Thus, the July mean SAT bias is the lowest at FI-Hyy due to a compensation between negative and positive biases (magnitude < 1°C), slightly higher at CA-Man from less compensation between negative and positive biases, and highest at US-Ivo (Fig. 8a).

Similarly, the mean diurnal cycles in the surface turbulent and radiative fluxes provide some explanation for their mean monthly values. The mean diurnal maximum net radiation in RASM1 is similar (531 and 509 W m$^{-2}$, respectively) at CA-



Man and FI-Hyy (Fig. 10d,e), but the observed net radiation in the daytime is higher at CA-Man than at FI-Hyy (432 and 327 W m$^{-2}$, respectively). Thus, the July mean net radiation is less biased at CA-Man than at FI-Hyy even though the net radiation is in the middle of the reanalysis spread at both locations (Fig. 8b). The net radiation at US-Ivo is even lower (maximum observed net radiation of 280 W m$^{-2}$) due to the higher latitude, but the daytime maximum is biased as high here

(134 W m$^{-2}$) as at CA-Man (Fig. 10e). RASM1 LH and SH fluxes are biased high in July at all three of these locations (Fig. 9c,d), because, while they are biased low at night, they are much too high compared to observations in the daytime (Fig. 10g-l). As expected from the net radiation biases, the LH and SH maximum biases are highest at FI-Hyy, being higher than even the reanalysis spread (Fig. 10h,k). The simulated LH fluxes are generally above the observational interannual variability at CA-Man and FI-Hyy but about the observational interannual variability maximum at US-Ivo, whereas

simulated SH fluxes are generally above the observational interannual variability during the early part of the day until the diurnal maximum. Also, intriguingly, the net radiation and LH and SH fluxes in RASM1 and the reanalyses can be out of phase with the observations, i.e., the daily maximum comes earlier in the day, even though model SATs are in phase with observations.

In light of the problems in WZ13 over Greenland due to its use of CRU, we compare simulated temperature to in situ

observations from five Greenland automated weather stations in Fig. 11). The reanalyses generally encompass the in-situ observations at all sites, whereas WZ13 has much warmer temperatures than observed from October to March at all locations. This further confirms that the use of CRU introduced warm biases over Greenland in winter in WZ13. On the other hand, RASM1 generally compares well with observations at all locations except NGRIP (Fig. 11a) and Summit (Fig. 11c) where RASM1 is too cold. From July to August, RASM1 is too warm compared to observations and reanalyses at all

sites.

Snow is a very important component of the Arctic system. Newly fallen snow has a much higher albedo than bare ground or vegetation. Additionally, snow insulates the ground from the cold air above in the winter. We compare RASM1 snow depth to the upscaled in situ observations in Fig. 12. Upscaled snow depth is higher in the mountainous ALASKA SOUTH region than in the flatter ALASKA MID region with maximum snow depths of ~1200 mm and ~650 mm in these regions,

respectively. RASM1 snow depth is lower in both regions, but it is able to simulate a snow depth closer to the upscaled observations in the relatively flat ALASKA MID region than in the mountainous ALASKA SOUTH region (~250 mm and ~600 mm lower, respectively). Dawson et al. (2016) found this to be the case for National Centers for Environmental Prediction (NCEP) models as well. Snow melt is initiated earlier in RASM1 than in the upscaled observations by ~0.5 months in ALASKA MID and a full month in ALASKA SOUTH. This is similar to what Hamman et al. (2016) found in

their comparison with a remotely sensed snow cover dataset.

**3.3 Comparison to SHEBA observations over sea ice**

Some of the model land biases are derived from similar biases over the neighboring central Arctic Ocean (Cassano et al. 2017). Since there is not much global gridded or in situ data for the central Arctic, we choose to rely on surface observations



made during the year-long SHEBA field campaign (Fig. 13). Observed LH flux is near-zero in autumn but is a little higher in summer (Fig. 13a), while observed SH flux is near-zero throughout the year (Fig. 13b). The observed LH flux is less reliable, as the mean is based on only one location (at the central tower). RASM1 SH flux compares well with observations during autumn and winter, but is higher than the observational range from May to July (Fig. 13b). RASM1 compares better

to observations than the reanalyses, which are largely outside of the observational range (Fig. 14b).

The SW and LW radiation components are also compared. Downward incident SW radiation in RASM1 is within the small observational spread from October 1997 to July 1998. Upward SW radiation is also within the observational spread from autumn to spring but peaks too low and early. Downward incident SW radiation is too low in late summer (Fig. 13c,d). On the other hand, downward LW radiation in RASM1 is generally slightly lower than the observational spread in winter but

compares well to observed from March to August 1998 (Fig. 13e). Interestingly, simulated upward LW radiation is within the observational spread throughout the year (Fig. 13f). Here, we also find that the reanalyses do not necessarily fall within (totally or partially) the observational spread for upward SW and LW radiation for part of the year.

The CESM1 ensemble mean does not consistently fall within the observational spread for the turbulent fluxes and LW radiation, and the ensemble spread may only partially fall within the observational spread. Because of this, the comparison

to the observational spread is more relevant than a comparison to the CESM1 ensemble spread. Surprisingly, despite the biases in the SW radiation components in RASM1 in summer, the net SW radiation is within the observational spread in June, August, and September 1998. The CESM1 ensemble mean is slightly above the observational spread from July to October 1998, but the ensemble spread is partially within it (supporting information Fig. S7a). Interestingly too, all model and reanalysis net LW radiation mostly falls outside of the observational spread throughout the campaign (supporting

information Fig. S7b). Over ice, the surface energy balance dictates that the sum of the net radiation and turbulent heat fluxes be balanced by the conductive flux through the snow (Maykut and Untersteiner 1971; Maykut 1978). In winter, this means that the strong LW radiative cooling is due almost exclusively to this conductive heat flux, since the SW radiation and LH and SH fluxes are practically zero. The larger radiative cooling in the models and reanalyses suggest that they produce more conductive heat flux than observed in winter. Sturm et al. (2001) found that snow/ice interface temperatures implied

from SHEBA observations were often much warmer (as much as 15°C warmer) than surface temperatures. The larger simulated conductive heat fluxes (not shown) imply that modeled snow/ice interface temperatures are warmer than they are observed to be. Any small difference in sea ice concentration or snow thickness could impact these conductive heat fluxes. The exploration of this is beyond the scope of this paper but could be a focus of further research.

As would be expected from the upward LW radiation, surface temperature is generally within the observational spread

throughout the year (Fig. 14a). Reanalysis surface temperatures are generally too high in autumn and winter, in agreement with their upward LW radiation (Fig. 13f). Wind speed in RASM1 is too high from February to August (Fig. 13b). This may explain the model overestimate of LH and SH fluxes in summer. The reanalysis spread is partially outside of the observational spread for wind speed from February 1998 onward.



### 3.4 Comparison to ship cruise observations

As expected from the regional comparisons made above, RASM1's SSTs are slightly colder than ship observations during CATCH/FASTEX in the wintertime North Atlantic and slightly warmer during Moorings in the autumnal North Pacific (Table 1). The SATs and specific humidity are similarly biased. Wind speed during the Atlantic cruises is underestimated in RASM1, while it is overestimated during Moorings. The reanalysis spread includes the observed mean for CATCH/FASTEX but is too high in all but wind speed during Moorings.

These surface conditions result in SH and LH fluxes that are slightly overestimated in RASM1. On the other hand, the reanalyses can be less biased, and CESM1 is slightly higher than observed with the exception of LH flux during CATCH/FASTEX. The reason for the difference between RASM1 and CESM1 biases becomes clear when we look at a scatter plot of model fluxes to ship observed fluxes. Despite RASM1 mean fluxes being higher than observed, the spread in this model's SH fluxes, for instance, is better than CESM1's which produces nearly constant fluxes at ~50 W m$^{-2}$ (Fig. 14). Also, the linear regression line for RASM1 fluxes is almost the one-to-one line, whereas that of CESM1 is slightly negative. Therefore, the apparent better capability of CESM1 is an artifact resulting from the compensation between underestimated and overestimated fluxes. While these results may be influenced by the nearby boundary, these improved fluxes suggest that the higher resolution afforded by the regional atmosphere and ocean models or the change in model physics afforded by WRF may offer an improvement to air-sea fluxes over the sub-Arctic oceans considering that both RASM1 and CESM1 are utilizing the same ocean model (POP) with the same air-sea flux algorithm.

### 4 Conclusions

In this study, we evaluate the newly developed version 1 of the Regional Arctic System Model (RASM1), a fully coupled atmosphere-land-ocean-sea ice model for improved high-resolution simulation of climate in the northern high-latitude region. The model is run over a pan-Arctic domain with WRF for the atmosphere, VIC for the land surface, POP for the ocean, and CICE for simulating sea ice. The model simulation is evaluated by using a coarser resolution global model (CESM1) and the spread in recent reanalyses of similar resolution to RASM as baselines of performance.

Overall, precipitation is similarly simulated in RASM1 as in CESM1 and the reanalyses. RASM1 precipitation compares better to GPCP and CMAP in every river basin except the Amur. RASM1 precipitation using ERA-Interim for BCs and spectral nudging is remarkably similar to the reanalyses, but the change in simulated precipitation by switching to CFSR for BCs is generally < 1 mm day$^{-1}$ and mostly statistically insignificant. As WRF contains a suite of various boundary layer and convective parameterizations, parameterization choice may affect these results. In a previous baseline simulation, different boundary layer and convective parameterization schemes were used, producing slightly less precipitation over the domain (Cassano et al., 2017). The impact of resolution on the simulation of precipitation will be investigated with the incorporation of a 25-km grid in RASM2.




Snow in RASM1 is underestimated by both simulations but is better simulated with a higher annual maximum in the flatter box in central Alaska (ALASKA MID) than in the more mountainous southern Alaska (ALASKA SOUTH) box. This is consistent with what Dawson et al. (2016) found when using this same data to evaluate the NCEP models. Broxton et al. (2016) found that a version of VIC that utilized a snow elevation band parameterization simulated snow the best out of the

other land models used in the Global Land Data Assimilation System (GLDAS, Rodell et al., 2014). This parameterization is currently not used in RASM1 but is being explored for use in RASM2.

There are mean biases in RASM1's land surface air temperature resulting from biases in surface radiation. In winter, SAT is too cold over much of the land within the RASM domain. Cassano et al. (2017) suggest that this is due to cloud biases. This still cannot be confirmed here, as cloud variables were still unable to be included in the model output of these simulations.

Such problems with simulating clouds have been noted before (e.g., Bromwich et al., 2009b; Porter et al., 2011; Bromwich et al., 2016). Clouds and their effect on interface processes might be better simulated if a newer version of WRF would have been used. Notably, WRF version 3.2 as is used in RASM1 neglects the radiative impacts of convective clouds. This effect will be incorporated with the inclusion of the latest version of WRF in RASM2.

The above monthly mean biases are a result of biases evident in the monthly mean diurnal cycles. For example, the monthly

mean RASM1 warm SAT biases in July are derived from more prominent warm biases during the day or consistently warmer SATs throughout the diurnal cycle. The surface turbulent flux and radiation are also similarly biased diurnally. Therefore, the key to advancing the simulation of SAT and the surface energy budget would be to improve the representation of the diurnal cycle of radiative and turbulent fluxes. The upcoming inclusion of WRF 3.2 may alleviate some of these diurnal cycle biases.

The comparison to the SHEBA observations from October 1997-October 1998 reveals that the reanalyses and the CESM1 ensemble spread do not always fall within the observational uncertainty. Therefore, the RASM1 comparison to the observational uncertainty is a better baseline in this instance. The surface temperature generally falls within the observational uncertainty for most months consistent with the upward longwave radiation. However, RASM1 wind speed is above the observational uncertainty during the spring and summer which may help to explain why the simulated latent and

sensible heat fluxes are biased high in summer.

An advantage of using RASM1 is that it captures the interannual and interdecadal variability in the climate of the Arctic region of which global models like CESM cannot do. This is shown in Fig. 16 for the SATs averaged over the central Arctic region, defined as 70°N and poleward. The RASM1 annual means (red dots) mimic the year-to-year variability of reanalysis annual means (black dots) despite being consistently lower than the reanalysis annual means. The underestimation in the

annual mean is due to the too cold SATs in winter as compared to reanalyses. On the other hand, the CESM1 annual means (green dots) mainly capture the overall increasing trend ($0.09$ K yr$^{-1}$) in regional temperatures since 1993 which is more representative of the reanalysis trend of $0.08$ K yr$^{-1}$ than RASM1 ($0.004$ K yr$^{-1}$). An ESM like CESM1 is not expected to capture the observed interannual and interdecadal variability, because its upper atmosphere is not nudged like RASM's. Also, it has to be initialized from arbitrary spun-up conditions and has no boundary conditions to constrain it. CESM1 is too



cold in winter but is also too cold in summer, whereas RASM1 compares well to the reanalyses in summer (supporting information Fig. S5). Since interannual and interdecadal variability is an important component of Arctic climate (Moritz et al., 2002; Stroeve et al., 2007), this represents an improvement in the simulation of the climate system of this region in RASM1 despite having mean biases. These biases exist at the surface despite being forced by external boundary conditions

and being nudged from the top, confirming that the fixing of internal problems in the model are important. These biases are a focus for further development of RASM version 2.

**Data availability.** The RASM output is archived at the U.S. DoD HPCMP.

**Code availability.** The RASM1 model code is archived on the Subversion server at the Naval Postgraduate School (https://svn.nps.edu) and cannot be publicly available due to copyright restrictions at this time. Access may be granted by

contacting Wieslaw Maslowski (maslowsk@nps.edu).

**Competing interests.** The authors have no competing interests.

**Acknowledgments.** This multi-institutional work was funded by the U.S. Department of Energy (DE-SC0006693, DE-SC0006178, DE-SC0006643, DE-FG02-07ER64460, DE-SC0006856, DE-SC0005783, and DE-SC0005522), by the U.S. National Science Foundation (PLR-1107788, PLR-1417818, and ARC1023369), and by the National Aeronautics and Space

Administration (NNX14AM02G). Computing resources were provided via a Challenge Grant from the U.S. Department of Defense (DoD) High Performance Computing Modernization Program (HPCMP). MERRA-2 data were downloaded from the Goddard Earth Sciences Data and Information Services Center (https://disc.sci.gsfc.nasa.gov). ERA-Interim 3-hourly and monthly mean data was downloaded from NCAR's Research Data Archive (RDA, http://rda.ucar.edu), while the synoptic hourly data was downloaded from ECMWF (http://apps.ecmwf.int/datasets/data/interim-full-mnth/levtype=sfc).

CFSR data was downloaded from NOAA National Centers for Environmental Information National Operational Model Archive and Distribution System [https://www.ncdc.noaa.gov/data-access/model-data/model-datasets/climate-forecast-system-version2-cfsv2#CFS Reanalysis (CFSR)]. WZ13 land SATs were downloaded from RDA. HadSST data is available from the UK Met Office. NOAA sea ice concentration CDR was downloaded from the National Snow and Ice Data Center (http://nsidc.org/data/g02202). CERES EBAF surface radiation was downloaded from the CERES web site

(https://ceres.larc.nasa.gov/order_data.php). CMAP was downloaded from NOAA's Earth System Research Laboratory (https://www.esrl.noaa.gov/psd/data/gridded/data.cmap.html). The ASTER GDEM v2 data used in the upscaling of in situ snow observations is available through the Data Pool at the NASA Land Processes Distributed Active Archive Center (LPDAAC). GC-Net weather station data were obtained from the Cooperative Institute for Research in Environmental Sciences (http://cires1.colorado.edu/steffen/gcnet/). FLUXNET tower data were downloaded from the FLUXNET web site

(http://fluxnet.ornl.gov).





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





**Table 1.** Comparison of Cruise Mean Observations and Simulated Values of Surface Meteorology and Turbulent Fluxes.

| | CATCH/FASTEX | Moorings |
|---|---|---|
| *SST (°C)* | | |
| Observations | 10.11 | 9.68 |
| RASM1 | 9.02 | 10.79 |
| Reanalyses | 9.67-10.14 | 14.37-14.50 |
| *SAT (°C)* | | |
| Observations | 7.15 | 8.37 |
| RASM1 | 6.09 | 10.37 |
| Reanalyses | 6.78-7.61 | 13.33-13.88 |
| *2-m specific humidity (g kg$^{-1}$)* | | |
| Observations | 4.85 | 5.90 |
| RASM1 | 4.78 | 6.47 |
| Reanalyses | 4.69-5.39 | 8.53-9.12 |
| *Wind speed (m s$^{-1}$)* | | |
| Observations | 10.63 | 4.98 |
| RASM1 | 9.64 | 8.20 |
| Reanalyses | 10.26-11.88 | 4.74-5.64 |
| *Sensible heat flux (W m$^{-2}$)[*]* | | |
| Observations | 62.68 | 2.11 |
| RASM1 | 70.58 | 17.71 |
| CESM1 | 50.77 | 54.34 |
| Reanalyses | 22.60-47.88 | 7.82-9.36 |
| *Latent heat flux (W m$^{-2}$)[*]* | | |
| Observations | 108.63 | 33.39 |
| RASM1 | 114.71 | 87.18 |
| CESM1 | 117.93 | 74.49 |
| Reanalyses | 96.99-124.96 | 39.42-49.83 |

[*]Sensible and latent heat fluxes are defined as positive upward.



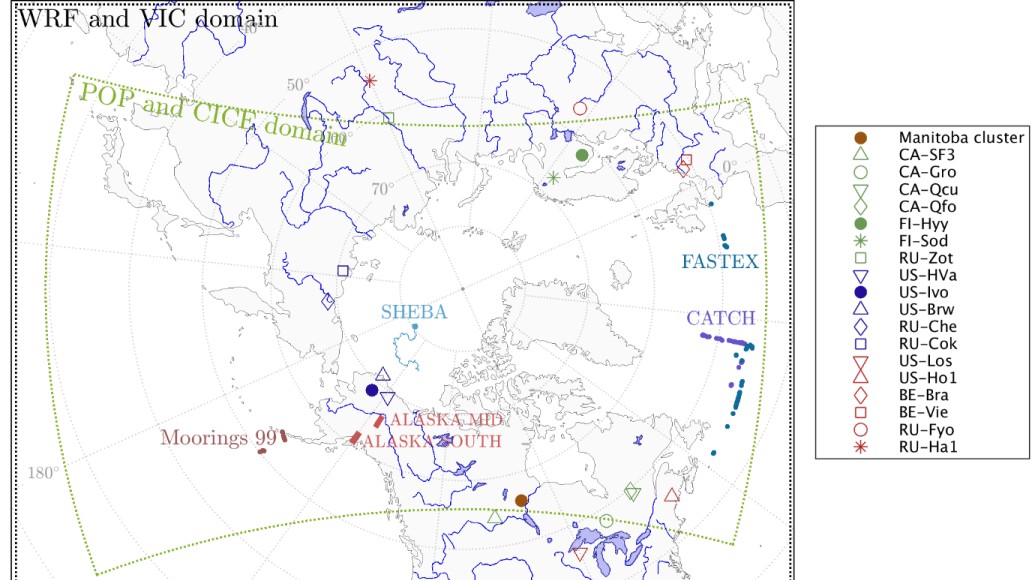

**Figure 1.** The RASM1 domains for the atmosphere (WRF) and land models (VIC) and for ocean (POP) and sea ice (CICE) models. The tracks of the ocean ship cruises (Moorings 99, CATCH, and FASTEX) and SHEBA are included. The location of the flux towers used in this study are also indicated by the symbols in the legend. The solid brown circle indicate the location of the Manitoba cluster and the three other towers that are shown in Figs. 8 and 11. The other symbols are the other flux towers. The two regions for snow depth evaluation with upscaled surface observations are also demarcated.

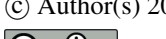



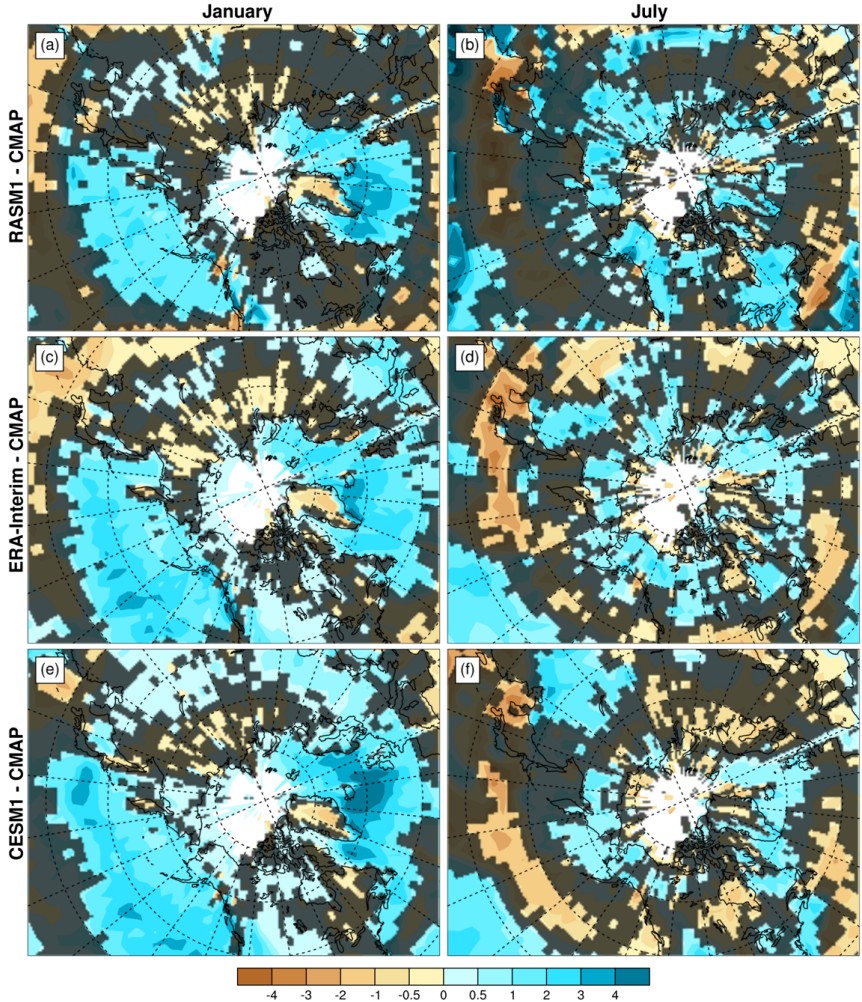

5 **Figure 2.** The bias in precipitation rate (mm day$^{-1}$) in (a,b) RASM1, (c,d) ERA-Interim, and (e,f) CESM1 from that of CMAP in January (left) and July (right) for 1990-2009. White areas are those in which CMAP contains missing data. The shading indicates grid cells with differences that are not statistically significant at the 95% level according to the Welch's two-sided t-test.




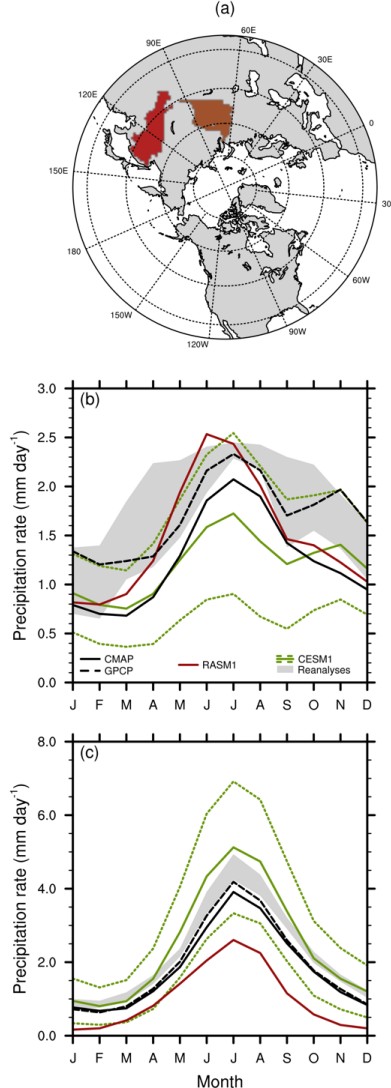

**Figure 3.** (a) The Ob (brown) and Amur (red) River basins. (bottom) Regional mean precipitation from CMAP (solid black), GPCP (dashed black), RASM1 (red), and CESM1 (green) along with the reanalysis spread (gray shading) over the Ob basin in (b) and the Amur basin in (c). The green dotted lines surrounding CESM1 indicates the ensemble maximum and minimum values.





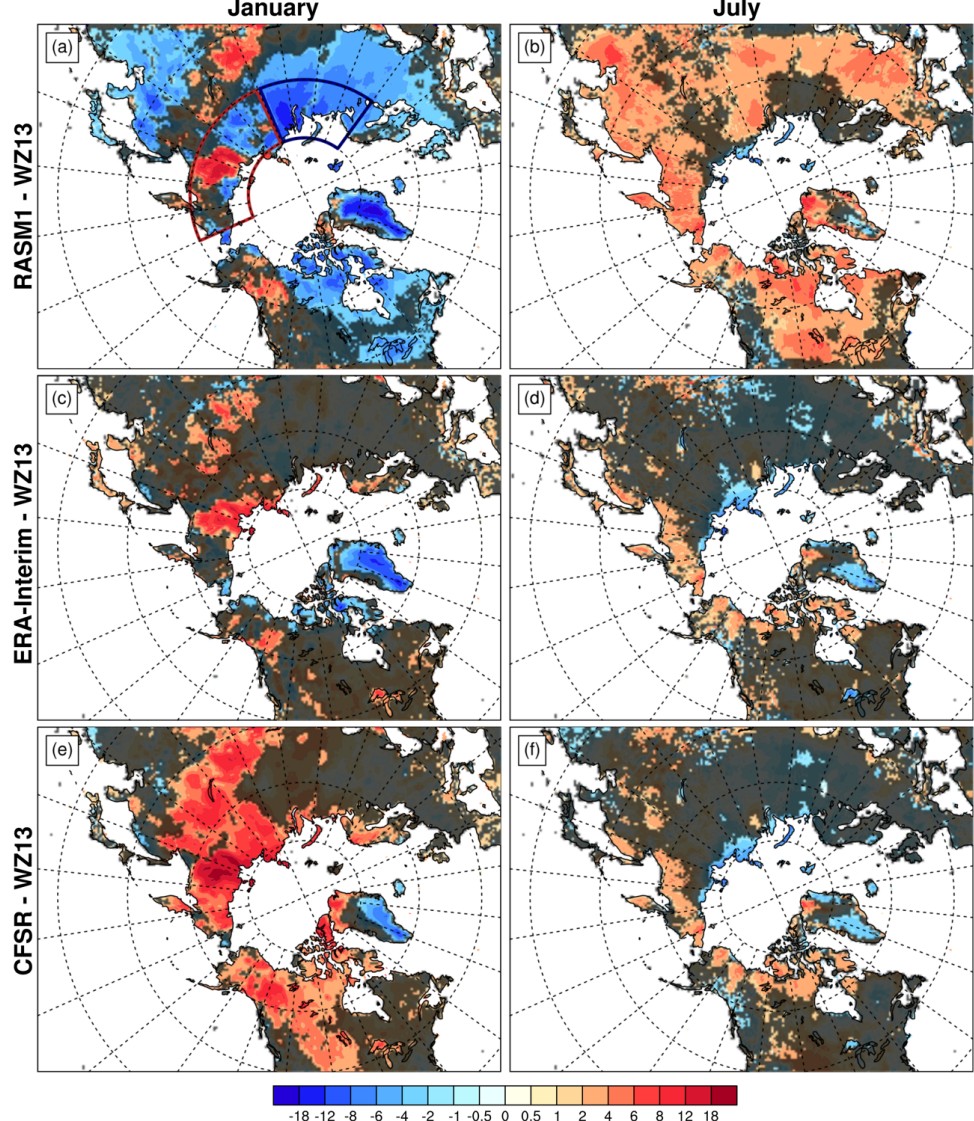

**Figure 4.** The bias in 2-m surface air temperature (SAT, °C) in (a,b) RASM1, (c,d) ERA-Interim, and (e,f) CFSR from that of the Wang and Zeng (2013) dataset in January (left) and July (right) for 1990-2009. The shading indicates differences that are not statistically significant at the 95% level according to the Welch's two-sided t-test. The blue and red boxes in panel (a) define the regions for which averages are produced in Fig. 7.




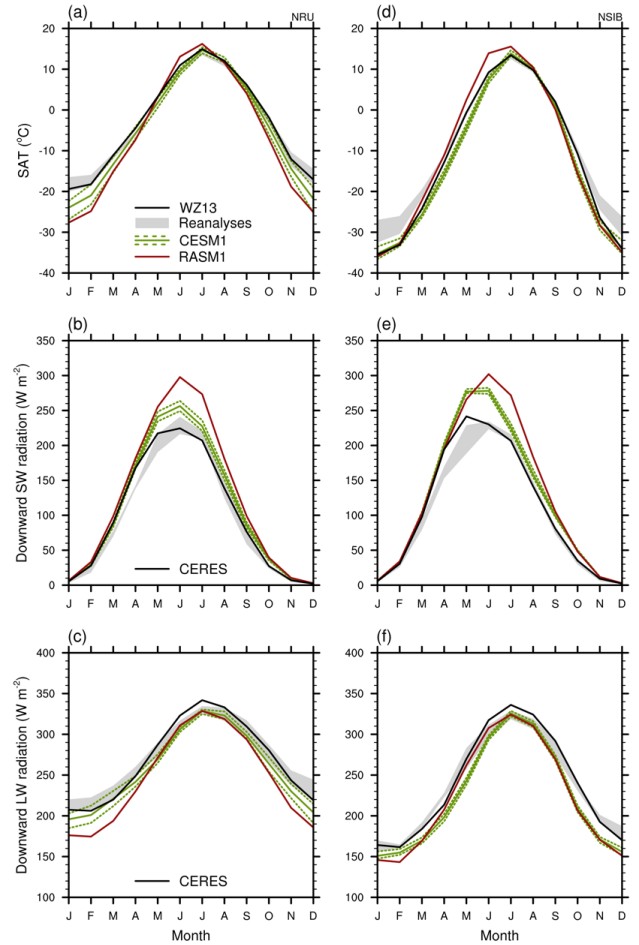

**Figure 5.** Regional mean of (a,d) 2-m air temperatures (SAT) and (b,e) surface incident downward shortwave (SW) radiation and (c,f) longwave (LW) radiation for (a-c) the NRU region defined in the blue box in Fig. 6a and (d-f) for the NSIB region defined in the red box in Figure 6a. Means are given for global datasets [Wang and Zeng (2013) SAT and CERES radiation, black], RASM1 (red), and CESM1 (green) along with the range in the three reanalyses (MERRA, ERA-Interim, and CFSR) indicated by the gray shading. The green dotted lines surrounding CESM1 indicates the ensemble maximum and minimum values.



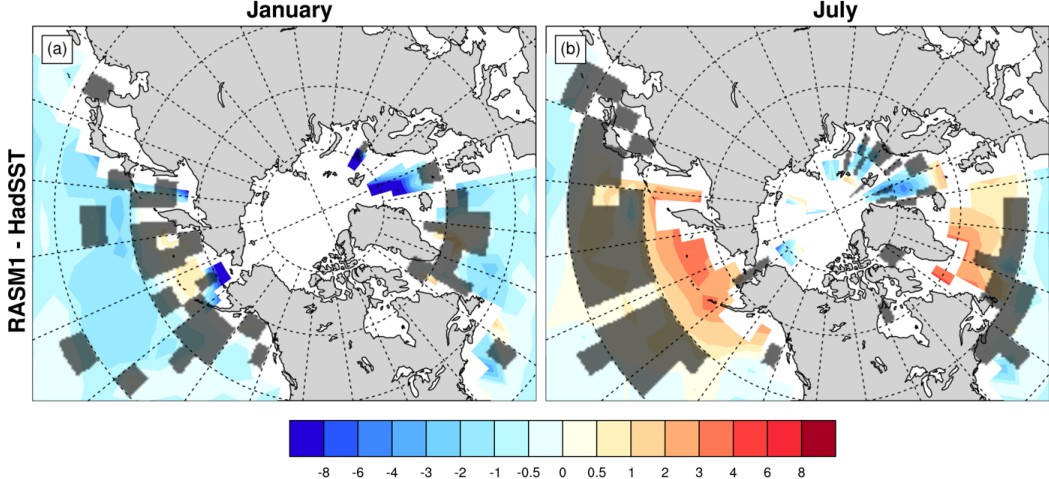

**Figure 6.** The bias in sea surface temperature (°C) in RASM1 from that of HadSST in (a) January and (b) July for 1990-
2009. The shading indicates differences that are not statistically significant at the 95% level according to the Welch's two-
sided t-test. The white grid cells are those that have missing data in HadSST.





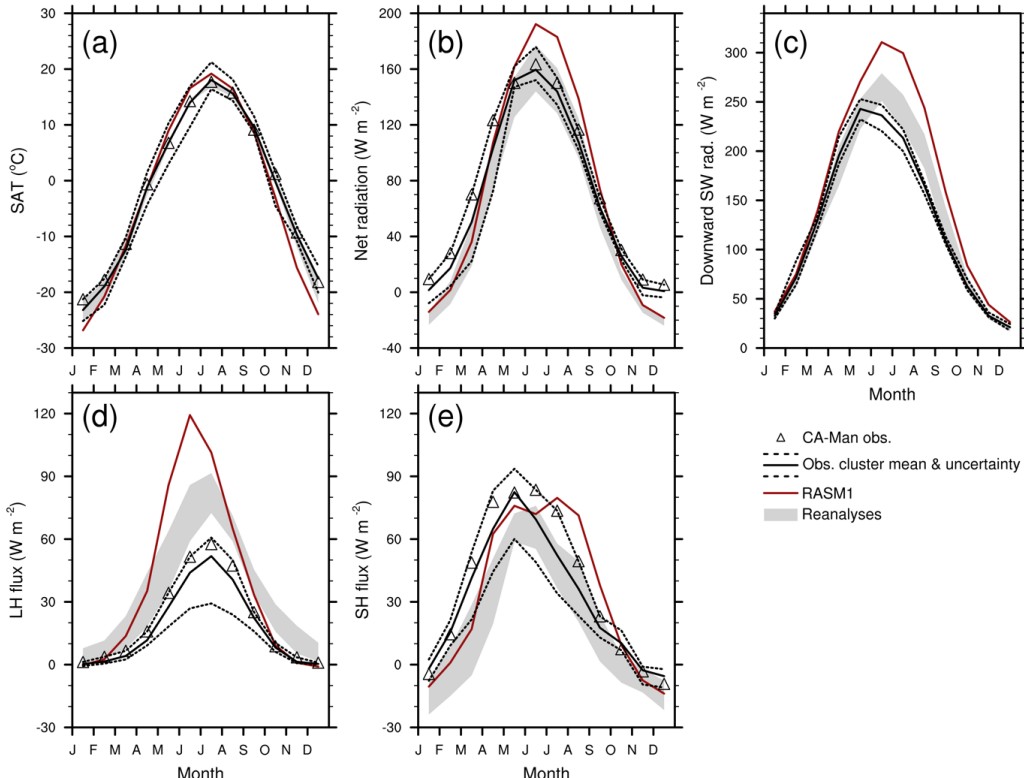

**Figure 7.** Mean annual cycle in (a) SAT, (b) net radiation, (c) downward incident shortwave (SW) radiation, (d) latent heat (LH) flux, and (e) sensible heat (SH) flux from land flux tower observations from the northern Manitoba cluster, RASM1 (red), and reanalyses. The cluster mean and spread (station minimum to maximum) is given as the solid and dotted black lines, respectively, while that of the individual tower CA-Man is given as the black triangles. The spread in the reanalyses is indicated by the gray shading.



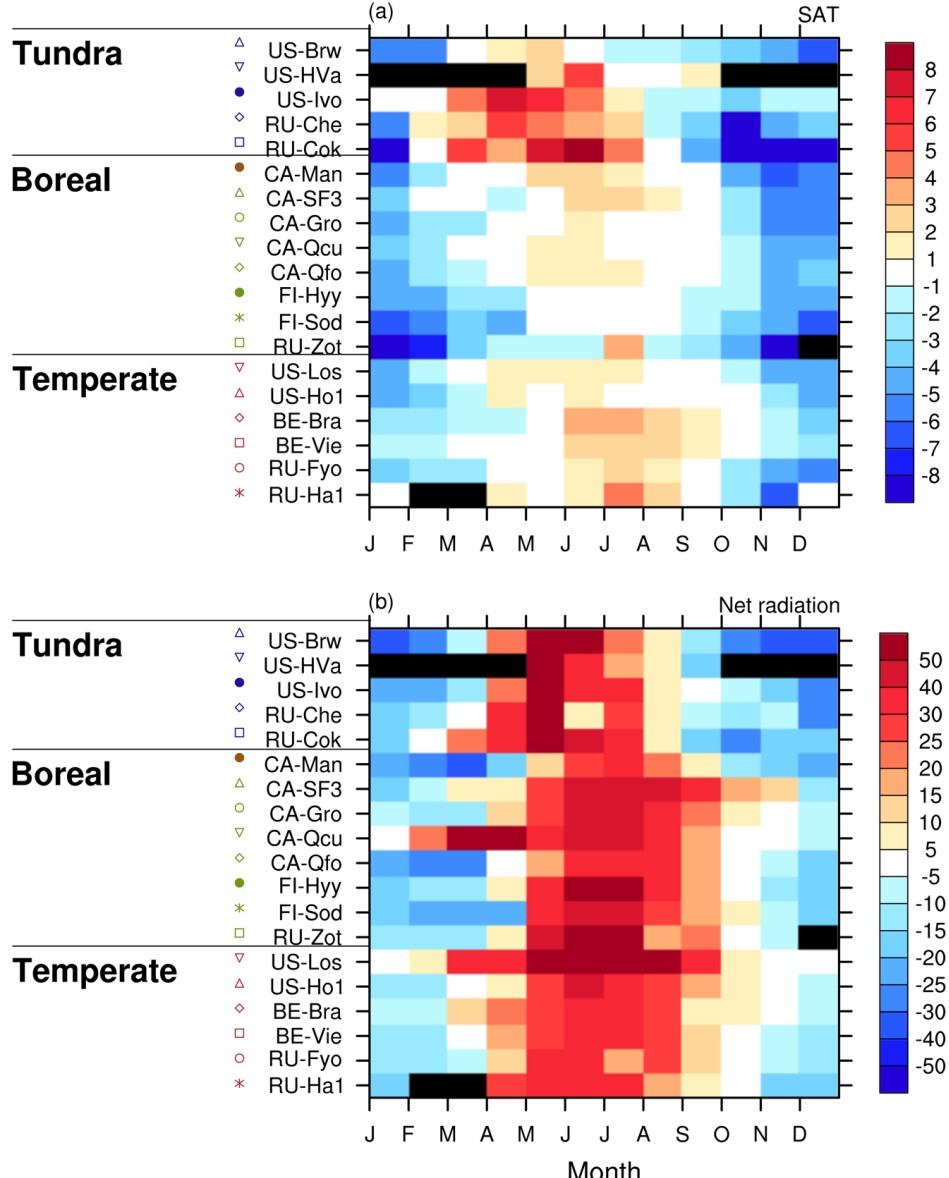

**Figure 8.** Monthly biases in RASM1 from flux tower observations for (a) SAT (°C) and (b) net radiation (W m$^{-2}$) at all locations CA-Man and all other locations outside of the Manitoba cluster. The black color indicates months with no data for that month at that station.



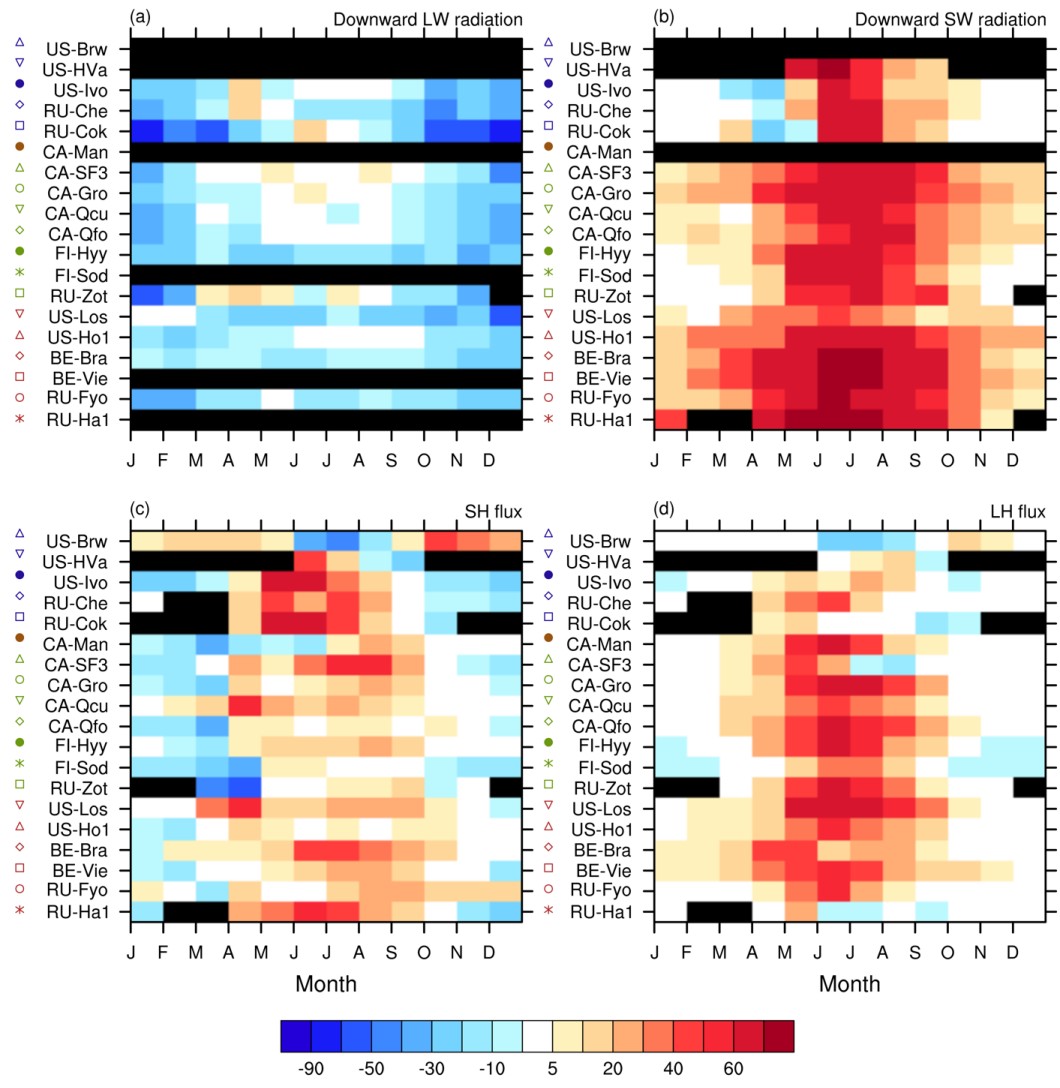

**Figure 9.** Monthly biases in RASM1 from flux tower observations for downward incident (a) SW and (b) LW radiation and (c) SH and (d) LH fluxes (W m$^{-2}$) at all locations outside of the Manitoba cluster and CA-Man. Positive SH and LH flux biases indicate more upward or less downward fluxes. The black color indicates months with no data for that month at that station.





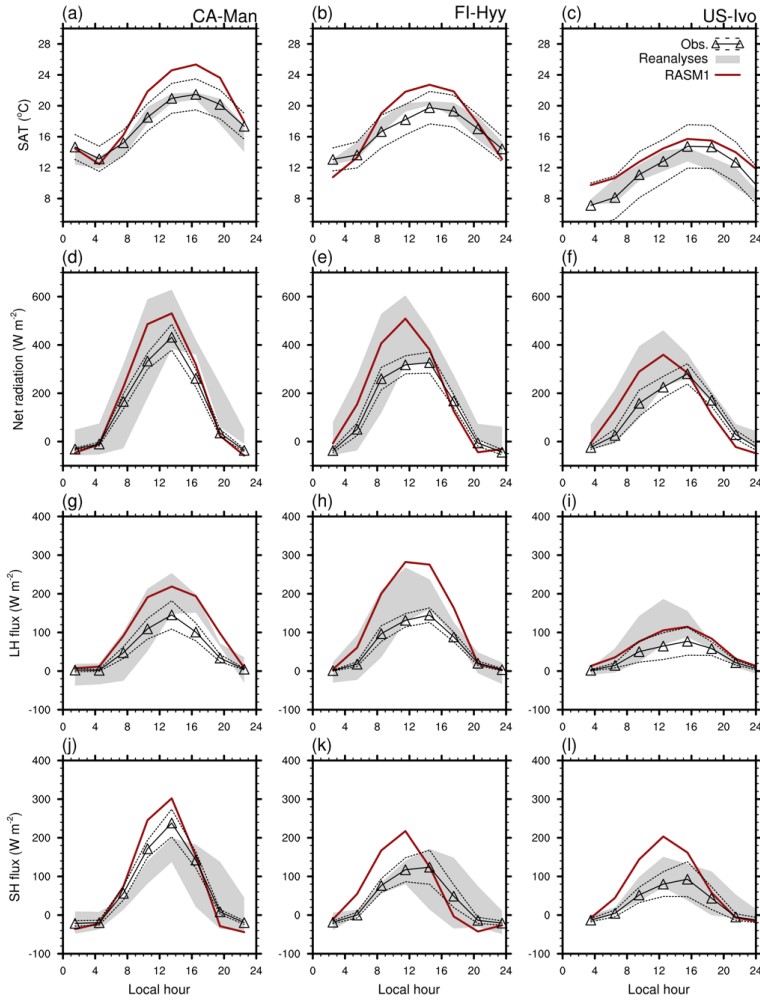

**Figure 10.** Mean diurnal cycles for July in (a-c) 2-m surface air temperature (SAT), (d-f) net radiation, (g-i) latent heat (LH) flux, and (j-l) sensible heat (SH) flux from flux tower observations (black), RASM1 (red), and RASM1a (purple), and reanalyses (spread shown as gray shading) at CA-Man (left column), FI-Hyy (center column), and US-Ivo (right column). The black dotted lines about the observation line represent the interannual variability ($\pm1\sigma$) in the observations.



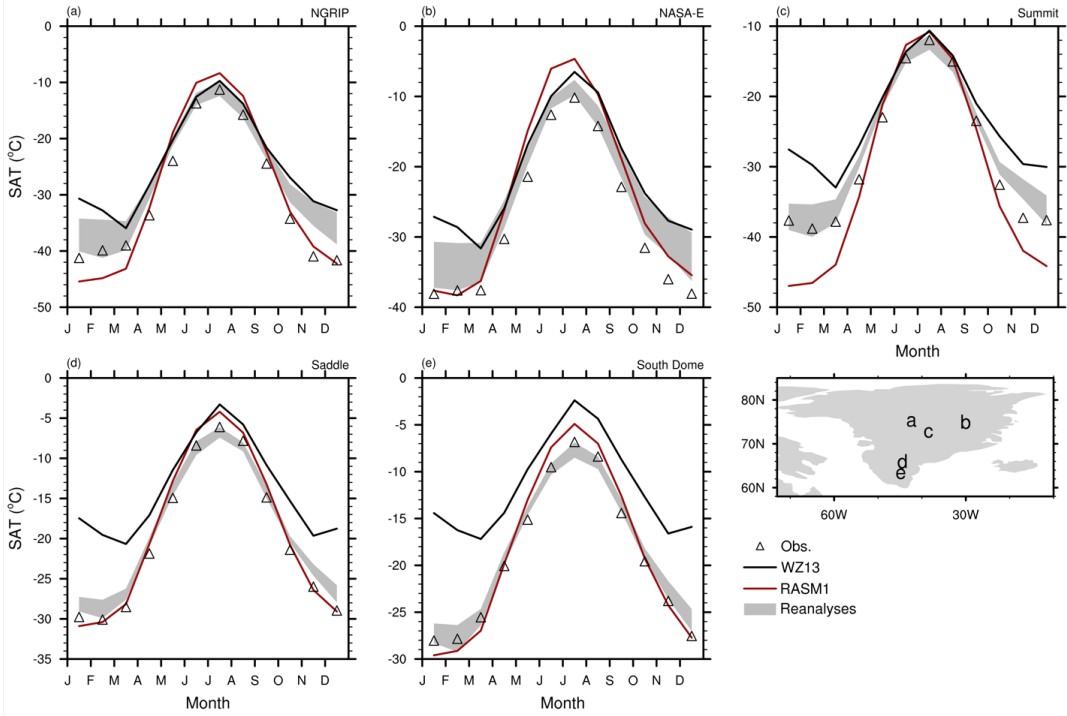

**Figure 11.** The mean annual cycle in SAT observed at automated weather stations across Greenland along with those from the Wang and Zeng (2013, WZ13) dataset (black), RASM1 (red), and the spread in the reanalyses indicated by the gray shading: at (a) NGRIP, (b) NASA-E, (c) Summit, (d) Saddle, and (e) South Dome. The locations of these sites are indicated in the map of Greenland.



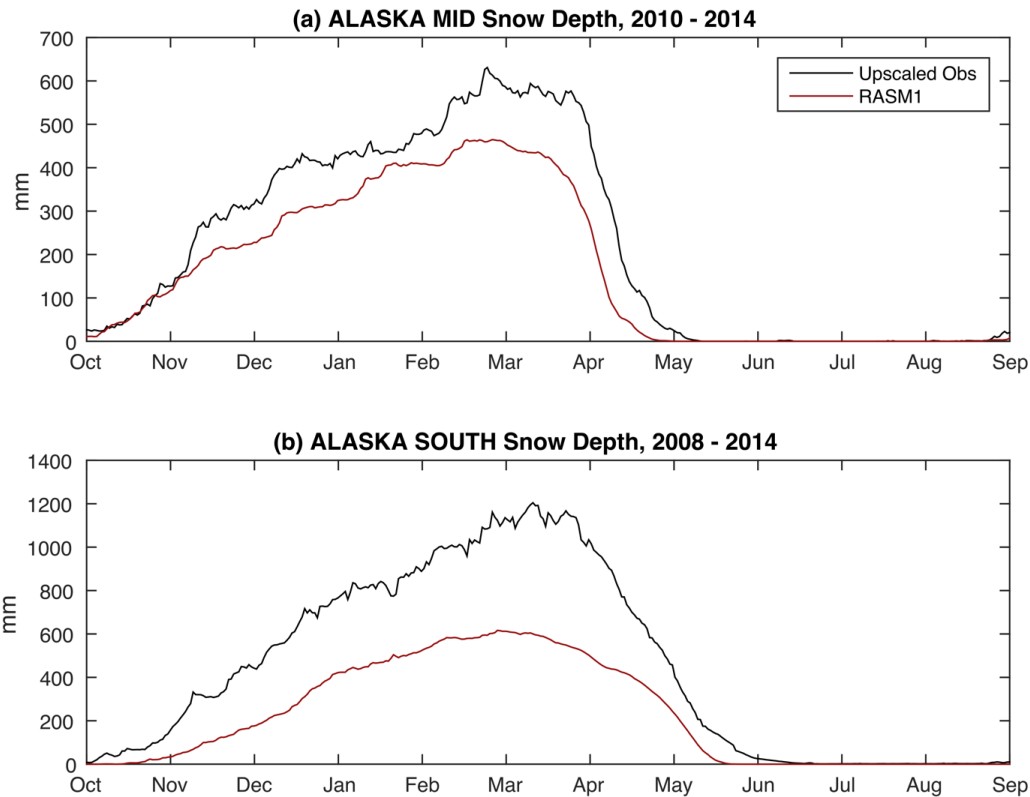

**Figure 12.** The mean annual cycle over a water year (October-September) of snow depth averaged for the 2° × 2° boxes defined in Fig. 1, (a) the middle of Alaska (ALASKA MID, top) and (b) southern Alaska (ALASKA SOUTH, bottom) as in upscaled observations (black) and in RASM1 (red).





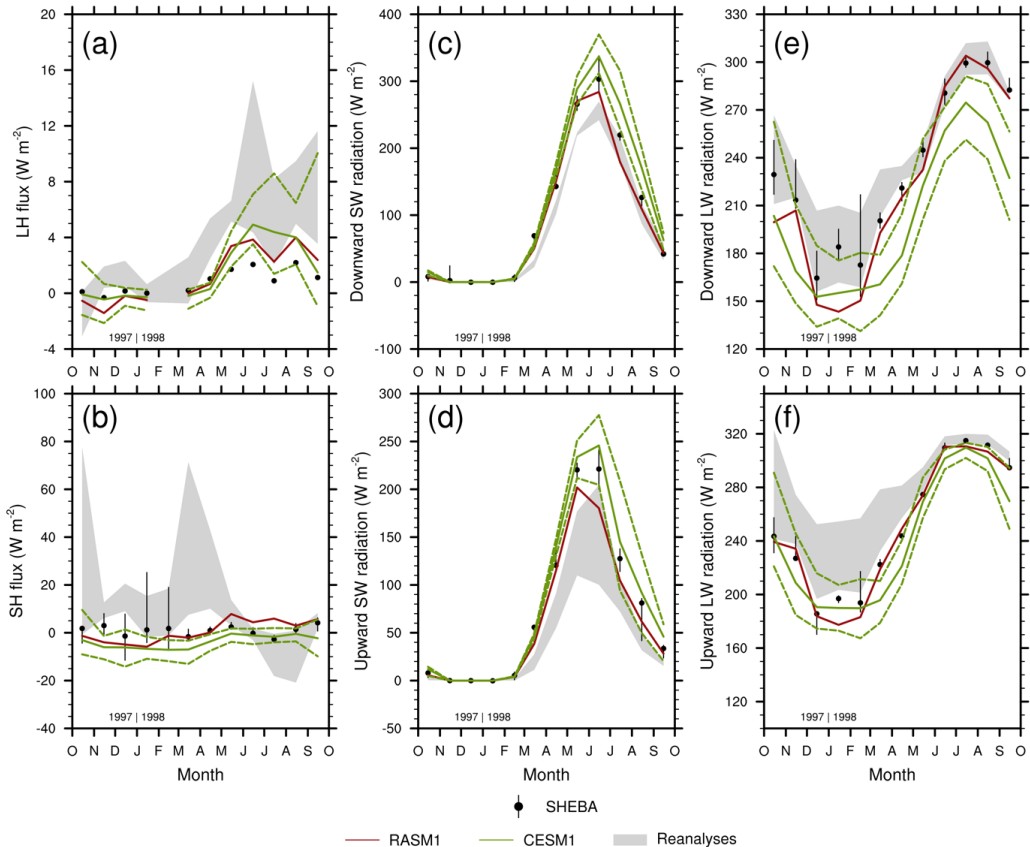

**Figure 13.** Comparison of monthly mean (a) latent heat (LH) flux, (b) sensible heat (SH) flux, (c) downward incident shortwave (SW) radiation, (d) upward reflected SW radiation, (e) downward incident longwave (LW) radiation, and (f) upward emitted LW radiation from SHEBA observations (black) with RASM1 (red), CESM1 (green), and reanalyses (gray shading). Positive LH and SH fluxes indicate upward fluxes. Observational spread is indicated by the vertical black lines extending from the circles, and the spread in reanalyses is shown by the gray shading. The green dashed lines surrounding CESM1 indicates the ensemble range (minimum to maximum).

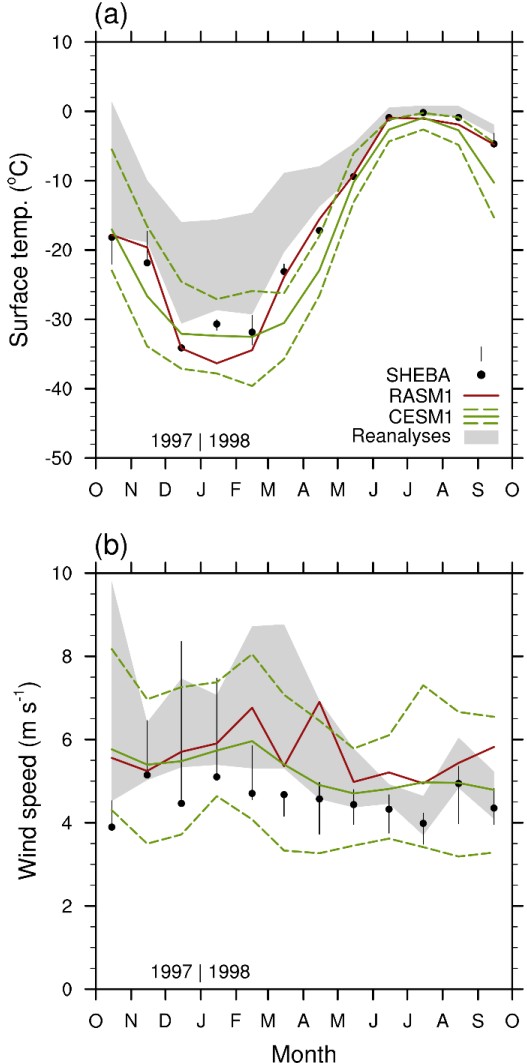

**Figure 14.** Comparison of monthly mean (a) surface temperature and (b) wind speed from SHEBA observations (black) with RASM1 (red), RASM1a (purple), CESM1 (green), and reanalyses (gray shading). Observational spread is indicated by the vertical lines extending from the circles, and the spread in reanalyses is shown by the gray shading. The green dashed lines surrounding CESM1 indicates the ensemble range (minimum to maximum).

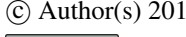



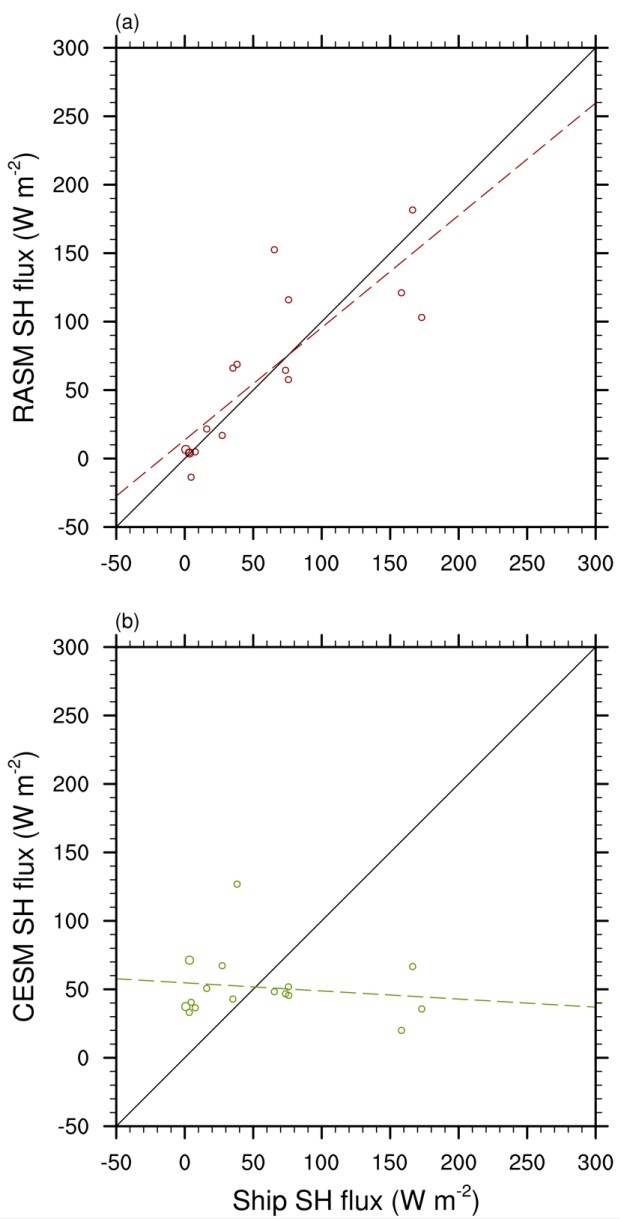

**Figure 15.** Daily mean model [(a) RASM1 and (b) CESM1] SH fluxes compared to corresponding observed fluxes aboard ship cruises in the North Pacific and Atlantic. The one-to-one line is indicated by the black line, and the dashed lines are the linear regressions of the model to ship fluxes.



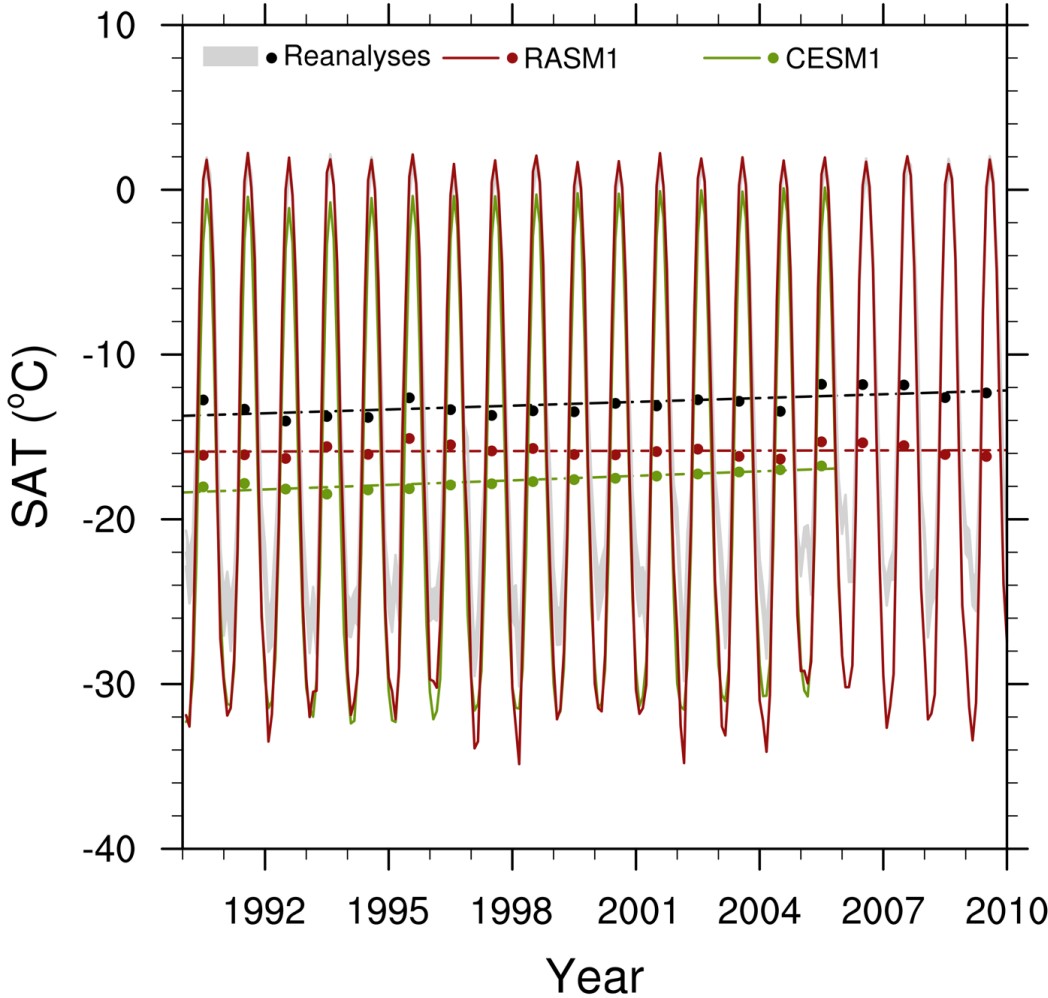

**Figure 16.** Monthly (lines and shading) and annual (dots) mean 2-m surface air temperature (SAT) over the central Arctic (70°N and poleward) in RASM1 (red), CESM (green), and the reanalyses (spread as gray shading and the annual mean as the black dots). The linear regressed trends in the annual SAT are shown as the dashed lines.