# Peer review of "Evaluation of the atmosphere-land-ocean-sea ice interface processes in the Regional Arctic System Model Version 1 (RASM1) using local and globally gridded observations"

_Geoscientific Model Development, 2018_

## Referee Comment (RC1) · Anonymous Referee #1 · 16 Jul 2018

Overall, I think the content of the paper is fine. I have a few minor suggestions for improving some of the details. My major quibble with the paper is the presentation: most of the paper is a recitation of similarities and differences among various simulations and data sets, which could be useful in and of itself to other users of the RASM model, but the paper was quite uninteresting to me until I read the Conclusions section where, finally, there was some indication of what scientific interest the paper might hold. I would prefer to see these things laid out earlier and/or more explicitly, as the take-home points, e.g. in the abstract.

For example, this paper compares not only RASM1, but also several different reanalysis data sets (which are also model output) to observed data. The results of the reanalysis-data comparison are interesting. Have other authors already made these comparisons? If so, cite them, e.g. in the first paragraph of section 3.2.

The insight gained from looking at diurnal cycles is another interesting aspect of the paper. It is mentioned in the abstract, but what is interesting about it is not there, only that it is related to the biases. Explaining a bit more would make the paper more compelling, so readers might actually plow through the recitation of similarities and differences.

I would like to see this in the abstract and/or introduction: What did you learn from this exercise that is new and applicable more generally than just this RASM model? Please make any new physical insights into the system the centerpiece of the paper, rather than just saying that it's a comparison of RASM with data and some other models.

More specific comments:

Abstract: In my opinion, statements like "The possible reasons for this result are discussed" are wasted words. Why not put a summary of the reasons themselves in the abstract?

page 2 line 28: is a paper published in 2011 really suitable support for a claim that something "cannot be represented within the computational constraints of the current generation of ESMs"?

In the first sentence of section 3, land biases are blamed on Arctic Ocean (sea ice?) biases. Do you know that it goes that direction? Why wouldn't the biases in/over the ocean be attributable to land biases?

page 16 lines 28-29: why was a new baseline run done? Is this why? If so, was it worth it?

page 17 line 18: why might WRF 3.2 improve the situation?

page 17 lines 26 and following (the last paragraph of the paper) should be mentioned in the abstract.

The paper is pretty sloppy. For instance,

English grammar and spelling is quite poor in this manuscript. E.g. lines 3 and 4 on page 2: "Sea ice thickness also decreased along with the sea ice extent decline ... decreaces ..."

Acronyms are not defined, or are defined long after they are first used. E.g. GCM, ESM, SAT.

Figure numbers are wrong, e.g. page 9 line 12 should refer to fig 2, not 3; line 11 of page 16 should refer to fig 15, not 14; fig 6 in the fig 5 caption should be fig 4 (I think)

Fig 10 and 14 captions refer to a purple trace as RASM1a, which is not shown.

Why is there a break in the plotted data of panel a in fig. 13?

---

## Referee Comment (RC2) · Anonymous Referee #2 · 10 Aug 2018

Review of Brunke et al. submitted to Geoscientific Model Development

General: This manuscript is revised from a manuscript previously submitted to a journal. It's much improved from the earlier version. It's now close to being ready for publication. Reviewers are often critical of model evaluations for multi-component models, since it's difficult to do a comprehensive evaluation in a standard-size manuscript. Such papers can be piecemeal and incomplete. However, there a series of papers on RASM with more detailed evaluations of the model components, so general paper on RASM performance is appropriate here. The figures and figures-linkage to manuscript could

use a more careful editing.

Specific Comments.

(1) latter part of page 2. When RASM first appeared it the resolution advantage over global models was especially an advantage. Global models are now encroaching on mesoscale resolutions in some cases. Also, variable resolution global models (http://www.tropmet.res.in/introspect/presentation/16feb/skamarock.pdf) are becoming or will become rivals to regional climate models. This is a concern to be kept in mind for RASM.

(2) second paragraph of page 3. This section fulfills an important need to define how the success of RASM results is to be measured.

(3) page 4, line 11. "ramped upwards from zero" is confusing.

(4) page 9, line 12. Should be a reference to Figure 2.

(5) page 9, line 20. "regional river basins" seems to come out of nowhere.

(6) page 9, line 23. "The biases relative to GPCP are generally the opposite ..." Are you discussing Figure 3?

(7) The figure captions for Figures 4 and 5 are not consistent.

(8) page 11, line 3. I realize you wish to minimize the discussion of cloud microphysics in this paper. However, it would be a good idea to at least mention the microphysics scheme here. In the second paragraph on this page, it could be mentioned that in general it is difficult to represent Arctic clouds in numerical models (e.g., Vavrus 2004; J. Climate, 17(3), 603-615).

(9) page 13, line 1. typo "is" to "are"

(10) page 13, line 11 "near-zero observations" ?

(11) page 15, third paragraph. The correct value of conductive heat fluxes through

the pack ice for SHEBA and presumably the Arctic Ocean as a whole is difficult to know, and appears to vary considerably over small distances (Sturm et al. 2001; 33(1), 213-220).

(12) Figure 14. I don't see a purple line.

(13) page 16, line 11. Appears to be a reference to Figure 15.
* * *

---

## Author Comment (AC1) · 11 Sep 2018

**Reply to Referee 1's comments**

The referee comments are in Arial. The authors' responses are given following each comment in *Times New Roman Italic.*

*We thank the referee for your helpful comments. We recognize that such an exhaustive model evaluation as this can be very tedious. We feel that that cannot be helped, but we now highlight the unique results (i.e., the diurnal cycle evaluation and the differentiation of RASM1 from global climate and Earth system models like CESM by capturing the actual interannual variability) in the abstract to possibly make it more interesting for a general reader to tread through the other results. We have addressed all of your other comments as well below.*

Overall, I think the content of the paper is fine. I have a few minor suggestions for improving some of the details. My major quibble with the paper is the presentation: most of the paper is a recitation of similarities and differences among various simulations and data sets, which could be useful in and of itself to other users of the RASM model, but the paper was quite uninteresting to me until I read the Conclusions section where, finally, there was some indication of what scientific interest the paper might hold. I would prefer to see these things laid out earlier and/or more explicitly, as the take-home points, e.g. in the abstract.

*Response: This has been addressed in response to the suggestions to add to the abstract made below.*

For example, this paper compares not only RASM1, but also several different reanalysis data sets (which are also model output) to observed data. The results of the reanalysis-data comparison are interesting. Have other authors already made these comparisons? If so, cite them, e.g. in the first paragraph of section 3.2.

*Response: A number of studies have evaluated reanalyses by comparing them to in-situ surface observations, but that is not the emphasis of this paper. We add to Line 4 on page 13: "Reanalyses have been evaluated previously through comparisons to surface in-situ observations (e.g., Decker et al., 2012; Betts et al., 2006; Zhou and Wang, 2016; Du et al., 2018), but this is not the focus here. Instead, we assess whether or not the reanalysis spread is within the observational spread."*

The insight gained from looking at diurnal cycles is another interesting aspect of the paper. It is mentioned in the abstract, but what is interesting about it is not there, only that it is related to the biases. Explaining a bit more would make the paper more compelling, so readers might actually plow through the recitation of similarities and differences.

*Response: We add to the abstract at Line 28: "At some locations, a minimal monthly mean bias is shown to be due to the compensation of roughly equal but opposite biases between day- and nighttime, whereas this is not the case at locations where the monthly mean bias is higher in magnitude. The diurnal cycle biases are derived from errors in the diurnal cycle of the energy balance (radiative and turbulent flux) components. Therefore, the key to advancing the simulation of SAT and the surface energy budget would be to improve the representation of the diurnal cycle of radiative and turbulent fluxes."*

I would like to see this in the abstract and/or introduction: What did you learn from this exercise that is new and applicable more generally than just this RASM model? Please make any new physical insights into the system the centerpiece of the paper, rather than just saying that it's a comparison of RASM with data and some other models.

*Response: This has been addressed in the response to the above comment and the one below about mentioning the advantage of using RASM1 in the abstract.*

More specific comments:

Abstract: In my opinion, statements like "The possible reasons for this result are discussed" are wasted words. Why not put a summary of the reasons themselves in the abstract?

*Response: This sentence is deleted.*

page 2 line 28: is a paper published in 2011 really suitable support for a claim that something "cannot be represented within the computational constraints of the current generation of ESMs"?

*Response: We change this to "that may be better represented by a regional coupled model."*

In the first sentence of section 3, land biases are blamed on Arctic Ocean (sea ice?) biases. Do you know that it goes that direction? Why wouldn't the biases in/over the ocean be attributable to land biases?

*Response: We change this sentence to "Some of the model land biases are similar to biases over the neighboring central Arctic Ocean."*

page 16 lines 28-29: why was a new baseline run done? Is this why? If so, was it worth it?

*Response: The current baseline simulation incorporating different boundary layer and convection parameterizations (MYNN and KF, respectively, vs. YSU and Grell-Dévényi before) was developed to alleviate cold SST biases that it had in the subpolar North Pacific from there being too little or too optically thin clouds over that region. The use of different parameterizations was motivated from the findings of Jousse et al. (2016), and they did alleviate the SST biases there (see Cassano et al., 2017). We now mention the previously used parameterizations in that sentence and add this next sentence: "The switch to MYNN and KF as is used in the current baseline was shown by Jousse et al. (2016) to produce a more realistic boundary layer height, liquid water path, and downward shortwave radiation in stratocumulus typical of the sub-polar oceans where the previous baseline produced a large cold SST bias."*

page 17 line 18: why might WRF 3.2 improve the situation?

*Response: We change "WRF3.2" to "the latest version of WRF", because WRF3.2 is what is currently used in RASM1 now. It currently does not include the radiative impact of convective clouds. RASM2 will include a newer version which does consider this important effect. We further add, "as it will include at least the radiation changes mentioned above."*

page 17 lines 26 and following (the last paragraph of the paper) should be mentioned in the abstract.

*Response: We mention this now as the last sentence in the abstract: "Still, an advantage of RASM1 is that it captures the interannual and interdecadal variability in the climate of the Arctic region of which global models like CESM cannot do."*

The paper is pretty sloppy. For instance,

English grammar and spelling is quite poor in this manuscript. E.g. lines 3 and 4 on page 2: "Sea ice thickness also decreased along with the sea ice extent decline . . . decreaces . . ."

*Response:  We correct this and all other typos.*

Acronyms are not defined, or are defined long after they are first used. E.g. GCM, ESM, SAT.

*Response:  We define these at their first instance now.*

Figure numbers are wrong, e.g. page 9 line 12 should refer to fig 2, not 3; line 11 of page 16 should refer to fig 15, not 14; fig 6 in the fig 5 caption should be fig 4 (I think)

*Response:  We correct these.*

Fig 10 and 14 captions refer to a purple trace as RASM1a, which is not shown.

*Response:  We remove these.*

Why is there a break in the plotted data of panel a in fig. 13?

*Response:  We add to the figure caption:  "No LH flux measurements are available in February 1998."*

---

## Author Comment (AC2) · 11 Sep 2018

**Reply to Referee 2's comments**

The referee comments are in Arial. The authors' responses are given following each comment in *Times New Roman Italic.*

*We thank the referee for your helpful comments both here and to the previous submission to another journal. We address all of your comments below.*

General: This manuscript is revised from a manuscript previously submitted to a journal. It's much improved from the earlier version. It's now close to being ready for publication. Reviewers are often critical of model evaluations for multi-component models, since it's difficult to do a comprehensive evaluation in a standard-size manuscript. Such papers can be piecemeal and incomplete. However, there a series of papers on RASM with more detailed evaluations of the model components, so general paper on RASM performance is appropriate here. The figures and figures-linkage to manuscript could use a more careful editing.

*Response: This is done now.*

Specific Comments.

(1) latter part of page 2. When RASM first appeared it the resolution advantage over global models was especially an advantage. Global models are now encroaching on mesoscale resolutions in some cases. Also, variable resolution global models (http://www.tropmet.res.in/introspect/presentation/16feb/skamarock.pdf) are becoming or will become rivals to regional climate models. This is a concern to be kept in mind for RASM.

*Response: At Line 30, the sentence "Many physical and biogeochemical processes in the Arctic are contingent upon interfacial exchanges at fine spatial scales and short time and cannot be represented within the computation constraints of the current generation of ESMs" is changed to "Many physical and biogeochemical processes in the Arctic are contingent upon interfacial exchanges at fine spatial scales and short time scales that may be better represented by a regional coupled model."*

(2) second paragraph of page 3. This section fulfills an important need to define how the success of RASM results is to be measured.

*Response: Agreed.*

(3) page 4, line 11. "ramped upwards from zero" is confusing.

*Response: We change this to "starts from zero at ~540 hPa increasing in strength upwards from there."*

(4) page 9, line 12. Should be a reference to Figure 2.

*Response:  We fix this.*

(5) page 9, line 20. "regional river basins" seems to come out of nowhere.

*Response:  We change this to "all of the Arctic region's river basins."*

(6) page 9, line 23. "The biases relative to GPCP are generally the opposite ..." Are you discussing Figure 3?

*Response:  This is in reference to a figure that is not shown.  We add "(not shown)" to the end of this sentence.*

(7) The figure captions for Figures 4 and 5 are not consistent.

*Response:  Figures 4 and 5 are updated to refer correctly back to each other.*

(8) page 11, line 3. I realize you wish to minimize the discussion of cloud microphysics in this paper. However, it would be a good idea to at least mention the microphysics scheme here. In the second paragraph on this page, it could be mentioned that in general it is difficult to represent Arctic clouds in numerical models (e.g., Vavrus 2004; J. Climate, 17(3), 603-615).

*Response:  In Line 13, we now mention that the too little or too optically thin cloud could be due to "the Morrison et al. (2009) two-moment microphysics scheme not producing enough supercooled water in mixed-phase clouds."  Plus, we add at the end of that paragraph (at Line 22):  "It should be noted that Arctic clouds are generally difficult to represent in models (e.g., Vavrus, 2004)."*

(9) page 13, line 1. typo "is" to "are"

*Response:  We fix this.*

(10) page 13, line 11 "near-zero observations" ?

*Response:  We replace this with "observations of ~0 W m$^{-2}$."*

(11) page 15, third paragraph. The correct value of conductive heat fluxes through the pack ice for SHEBA and presumably the Arctic Ocean as a whole is difficult to know, and appears to vary considerably over small distances (Sturm et al. 2001; 33(1), 213-220).

*Response:  We now mention at Line 28 that "these conductive heat fluxes…can vary considerably over small distances (Sturm et al., 2001)."*

(12) Figure 14. I don't see a purple line.

*Response: The reference to the purple line is removed.*

(13) page 16, line 11. Appears to be a reference to Figure 15.

*Response: We correct this.*